# GSAlign: Geometric and Semantic Alignment Network for Aerial-Ground Person Re-Identification

**Qiao Li**[1], **Jie Li**[2],[*] **Yukang Zhang**[2], **Lei Tan**[3],[†]
**Jing Chen**[1], **Jiayi Ji**[2,3]
[1]Key Laboratory of Aerospace Information Security and Trusted Computing, Ministry of Education,
School of Cyber Science and Engineering, Wuhan University
[2]Xiamen University
[3]National University of Singapore
liqiaoqiao233@whu.edu.cn, lei.tan@nus.edu.sg

## Abstract

Aerial-Ground person re-identification (AG-ReID) is an emerging yet challenging task that aims to match pedestrian images captured from drastically different viewpoints, typically from unmanned aerial vehicles (UAVs) and ground-based surveillance cameras. The task poses significant challenges due to extreme viewpoint discrepancies, occlusions, and domain gaps between aerial and ground imagery. While prior works have made progress by learning cross-view representations, they remain limited in handling severe pose variations and spatial misalignment. To address these issues, we propose a Geometric and Semantic Alignment Network (GSAlign) tailored for AG-ReID. GSAlign introduces two key components to jointly tackle geometric distortion and semantic misalignment in aerial-ground matching: a Learnable Thin Plate Spline (LTPS) Module and a Dynamic Alignment Module (DAM). The LTPS module adaptively warps pedestrian features based on a set of learned keypoints, effectively compensating for geometric variations caused by extreme viewpoint changes. In parallel, the DAM estimates visibility-aware representation masks that highlight visible body regions at the semantic level, thereby alleviating the negative impact of occlusions and partial observations in cross-view correspondence. A comprehensive evaluation on CARGO with four matching protocols demonstrates the effectiveness of GSAlign, achieving significant improvements of +18.8% in mAP and +16.8% in Rank-1 accuracy over previous state-of-the-art methods on the aerial-ground setting.

## 1 Introduction

Person re-identification (ReID), aiming to address the problem of matching people over a distributed set of nonoverlapping cameras, has attracted intensive attention in the last few years due to its wide applications in surveillance systems. Although traditional person re-identification (ReID) has achieved remarkable progress [1, 2, 3, 4, 5], there is growing interest in aerial-ground ReID (AG-ReID) settings, driven by the increasing deployment of unmanned aerial vehicles (UAVs) and low-altitude platforms in surveillance applications [6, 7, 8, 9, 10]. This, in practice, puts the Re-ID problem in an aerial-ground setting and requires the approaches to properly handle both the significant geometric view-variation and semantic misalignment.

Despite its potential, AG-ReID remains a highly underexplored and technically challenging problem. Unlike conventional ReID tasks [11, 12, 13, 14, 15], AG-ReID suffers from extreme viewpoint

---

[*]Project Leader
[†]Corresponding Author: Lei Tan

39th Conference on Neural Information Processing Systems (NeurIPS 2025).

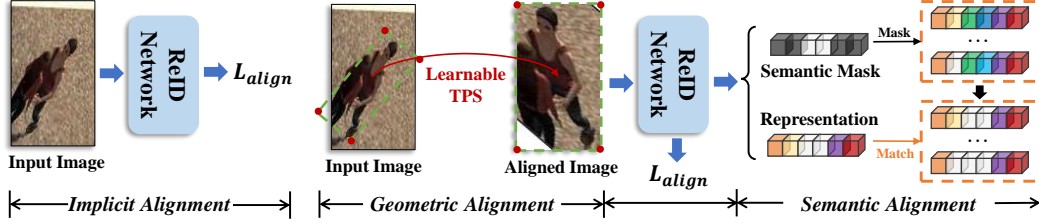

(a) Current alignment strategy.      (b) Our proposed Geometric and Semantic Alignment Network (GSAlign).

Figure 1: **Illustration of the motivation of GSAlign.** **(a)** Prior methods rely solely on implicit alignment, which is insufficient to fully address spatial and semantic distortions. **(b)** In contrast, our GSAlign performs explicit alignment at both the geometric and semantic levels via LTPS and visibility-aware semantic masks, respectively. This design equips GSAlign with a stronger capability for robust aerial-ground matching.

disparities, where aerial images exhibit severe top-down perspectives while ground views contain mostly frontal or profile views. These differences cause significant geometric distortions and drastic appearance changes, which conventional ReID models that are primarily trained on ground-view data are ill-equipped to handle. In addition, AG-ReID often involves frequent occlusions and unbalanced visibility across views, further complicating cross-view matching.

To mitigate these challenges, several recent studies have attempted to adapt existing ReID techniques to the aerial-ground scenario [16, 8]. These methods seek to learn modality-specific representations or map features into a shared embedding space using metric learning or adversarial objectives. Others employ cross-view decomposition strategies or design view-invariant constraints to bridge the modality gap. However, these methods tend to focus on global alignment and overlook two critical issues: (1) the severe geometric distortion induced by cross-view perspectives, and (2) semantic misalignment caused by partial occlusions and varying visible body regions. These limitations lead to suboptimal performance, especially under large viewpoint gaps.

In this paper, as shown in Figure 1 (b), we propose a Geometric and Semantic Alignment Network (GSAlign), which is specifically designed to tackle the core challenges of Aerial-Ground Re-Identification (AG-ReID). The key insight of our approach is to explicitly model both geometric deformation and visibility-aware semantics within a unified ViT framework. To this end, GSAlign introduces two complementary modules: First, the Learnable Thin Plate Spline (LTPS) Transformation Module. LTPS adaptively warps pedestrian representations based on a set of learned keypoints. Unlike hand-crafted alignment strategies, our LTPS module is fully differentiable and end-to-end trainable, enabling the network to learn viewpoint-conditioned transformations that effectively and dynamically compensate for severe spatial distortions caused by extreme cross-view differences. To avoid potential errors caused by one-shot correction, LTPS is progressively integrated into the hierarchical layers of a Vision Transformer (ViT). This design enables the network to iteratively refine geometric alignment across layers, allowing fine-grained, stage-wise correction throughout the feature propagation process. Second, the Dynamic Alignment Module (DAM). DAM enables each input image to predict its own visibility-aware representation masks at the semantic level, highlighting visible body regions while suppressing noisy or occluded areas. The predicted masks are then applied to the corresponding gallery features to filter out noisy signals from invisible regions, thereby enhancing cross-view feature alignment. By dynamically adapting to the visibility of different body parts, DAM guides the network to focus on semantically consistent and identity-discriminative cues across aerial and ground views. This allows the model to focus on semantically consistent cues across views, thereby improving robustness under occlusions and pose variations.

To sum up, the main contributions of this paper are as follows:

- We propose GSAlign, a novel framework for aerial-ground person re-identification that jointly addresses geometric deformation and semantic misalignment within a unified architecture. GSAlign is specifically designed to handle the extreme cross-view variations and visibility inconsistencies inherent in UAV-to-ground matching scenarios.

- We introduce a Learnable Thin Plate Spline (LTPS) Module and a Dynamic Alignment Module (DAM). LTPS performs keypoint-guided feature warping to compensate for severe spatial

distortions, while DAM enhances semantic alignment by estimating visibility-semantic representation masks to highlight visible body regions and suppress noisy or occluded areas.

- Extensive experiments on the challenging CARGO dataset validate the effectiveness of GSAlign, which achieves state-of-the-art performance with absolute gains of +18.8% in mAP and +16.8% in Rank-1 accuracy on the aerial-ground setting.

## 2 Related Work

Building upon well-established ground surveillance infrastructure, advanced person re-identification algorithms have made significant progress across a range of scenarios, including general scenarios [17, 18, 1, 4], occluded scenarios [19, 12, 20, 21], cross-modal scenarios [22, 23, 24, 25], multi-spectral scenarios [26, 27, 28], and cross-spectral scenarios [29, 30, 31, 32, 33].

However, with the proliferation of unmanned aerial vehicles (UAVs) and low-altitude platforms augmenting traditional surveillance systems, person ReID research is increasingly extending to aerial-view scenarios [34]. Initial efforts in this direction focused on aerial-aerial retrieval tasks using datasets such as UAV-Human [35] and PRAI-1581 [7]. More recently, attention has shifted towards connecting the aerial views with traditional ground systems. Aerial-Ground person ReID (AG-ReID), where the query and gallery images originate from fundamentally different viewpoints, typically aerial and ground-level perspectives. This setup introduces severe spatial and semantic discrepancies, often leading to substantial degradation in identity-preserving cues such as pose, silhouette, and clothing texture [36]. Several approaches have been proposed to mitigate these challenges. Nguyen *et al.* [6] proposed a dual-stream structure guided by attributes to enhance semantic disentanglement, later extended to a three-stream model with modality-aware supervision [16]. Zhang *et al.* [8] developed the View-Decoupled Transformer (VDT), which explicitly models and separates viewpoint-specific and viewpoint-invariant features. SD-ReID [37] proposes a generative framework that leverages diffusion models to synthesize view-specific features and enhance view-invariant person representations. In addition, LATex [38] incorporates attribute-based text knowledge via prompt-tuning strategies on vision-language models, explicitly exploiting the viewpoint-invariant nature of person attributes to improve cross-view retrieval. Despite these advances, existing AG-ReID methods still face difficulties in handling extreme spatial distortions and in leveraging visible semantic regions effectively. To overcome these limitations, we propose **GSAlign**, a novel framework that performs instance-adaptive geometric transformation and visibility-aware semantic alignment, promoting robust and spatially consistent feature learning across aerial and ground domains.

## 3 Methodology

### 3.1 Overview

The overall framework of the **Geometric and Semantic Alignment Network (GSAlign)**, illustrated in Fig. 2, is built upon a ViT-Base backbone and designed to address feature misalignment under complex multi-view conditions. Inspired by the success of the View-Decoupled Transformer (VDT) [39], we adopt the VDT architecture as the foundation of GSAlign, which effectively captures both global semantics and fine-grained structural cues. Following its design, we introduce an additional view token to model viewpoint-specific information. The token sequence during training is defined as:

$$[\mathbf{X}_{\text{cls}}, \mathbf{X}_{\text{view}}, \mathbf{X}_{\text{img}}] = \text{VDT}\left([\mathbf{X}_{\text{cls}}, \mathbf{X}_{\text{view}}, \text{tokenize}(\mathbf{X}_{\text{img}})]\right), \quad (1)$$

where $\mathbf{X}_{\text{cls}}$, $\mathbf{X}_{\text{view}}$, and $\mathbf{X}_{\text{img}}$ represent the class token, view token, and image tokens, respectively.

To improve geometric robustness, we integrate a **Learnable Thin Plate Spline module (LTPS)** into transformer layers. This module predicts rotation-aware control point offsets and applies a non-rigid spatial transformation to warp local patch features. By doing so, LTPS mitigates pose-induced geometric distortions and enhances structural alignment across views. In addition, we introduce a **Dynamic Alignment Module (DAM)** during training, which generates channel-wise semantic masks guided by class-specific prototype features. These masks dynamically highlight identity-relevant subspaces, allowing the model to focus on discriminative patterns while suppressing background noise and occlusion artifacts. The generated masks serve a specific purpose: aligning the features toward consistent subspaces. Overall, the GSAlign integrates the LTPS and DAM, jointly optimizing

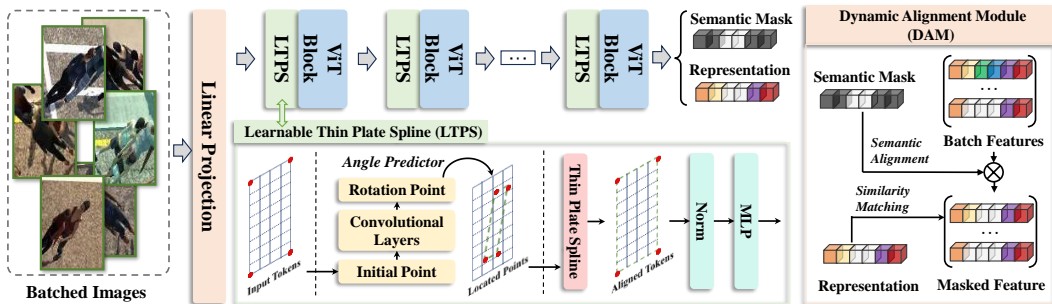

Figure 2: **Overview of the GSAlign architecture.** Given aerial or ground-view inputs, GSAlign first applies an initial geometric transformation via a Learnable Thin Plate Spline (LTPS) module, followed by progressive alignment through LTPS blocks inserted before each ViT layer. In parallel, a Dynamic Alignment Module (DAM) generates a visibility-aware semantic mask according to the input image, which is then applied to the representations of other images in the batch to suppress irrelevant or occluded features.

for classification accuracy, feature consistency, and robustness to view and shape variations. This synergy enables GSAlign to show a strong re-identification ability in severe view variations.

## 3.2 Learnable Thin Plate Spline

To address the spatial misalignment problem in person re-identification, we design a Learnable Thin Plate Spline (LTPS) deformation module that explicitly models geometric variations arising from changes in human pose and viewpoint. By dynamically predicting control point displacements and rotation angles, the module performs nonlinear transformations on feature maps, thereby enhancing their ability to capture non-rigid spatial deformations. As illustrated in Fig. 2, the LTPS modules are integrated into each transformer block to build a hierarchical deformation-aware representation. The LTPS modules at shallower layers focus on capturing local deformation details, while those at deeper layers are responsible for modeling global pose variations.

The core idea of the LTPS module is to perform non-rigid deformation based on control points interpolation. We initialize a set of 2D source contro points $\mathbf{P}_s \in \mathbb{R}^{K \times 2}$ as a regular grid uniformly distributed over the normalized coordinate space $[-1, 1] \times [-1, 1]$ and treat them as learnable parameters optimized during training. In parallel, we define a set of target control points $\mathbf{P}_t \in \mathbb{R}^{K \times 2}$ which are fixed and share the same regular grid positions as the initial $\mathbf{P}_s$, representing the canonical target shape. During training, a rotation angle is predicted from the input features and applied to $\mathbf{P}_s$, and a smooth deformation mapping is constructed by enforcing an interpolation constraint from the rotated $\mathbf{P}_s$ to the fixed $\mathbf{P}_t$, guiding the feature map to undergo spatial transformation.

In traditional thin plate spline (TPS) transformations, the source control points $\mathbf{P}_s$ are fixed, and spatial rearrangement is achieved only by learning the target control points $\mathbf{P}_t$, which limits the ability to handle global rotations and complex deformations. To address this, we design a rotation prediction module that predicts a rotation angle from the global orientation of the input feature map, and applies this rotation to the source control points for improved modeling of spatial deformations. The process is defined as:

$$\mathbf{P}_s^{(\text{rot})} = \mathbf{P} \begin{bmatrix} \cos\theta & -\sin\theta \\ \sin\theta & \cos\theta \end{bmatrix}^\top, \quad \theta = f_\theta(\mathbf{F}). \tag{2}$$

Here, $\mathbf{F}$ represents the patch-level feature map of each layer, and $f_\theta(\cdot)$ is the rotation prediction network that outputs the angle $\theta \in \left[-\frac{\pi}{2}, \frac{\pi}{2}\right]$.

To obtain the transformation function $\mathbf{T}(\cdot)$, the TPS requires the source control points $\mathbf{P}_s$ and the target control points $\mathbf{P}_t$ to form the following interpolation constraint:

$$\mathbf{T}(\mathbf{P}_{s,i}) = \mathbf{P}_{t,i}, \quad \forall i = 1, 2, 3, ..., K. \tag{3}$$

This constraint requires that the transformation function precisely maps the source points $\mathbf{P}_s$ to the target points $\mathbf{P}_t$ at each control point. Based on this, for the rotated source points $\mathbf{P}_s^{rot}$ and the target

control points $\mathbf{P}_t$, the TPS transformation function $\mathbf{T}(\cdot)$ is defined as follows:

$$\mathbf{T}(\mathbf{p}_x) = \mathbf{A} \cdot \mathbf{p}_x + \sum_{i=1}^{K} \mathbf{w}_i \cdot U(\|\mathbf{x} - \mathbf{P}_{s,i}^{(\text{rot})}\|), \tag{4}$$

where $\mathbf{p}_x$ can be any point from the feature map. $\mathbf{A} \in \mathbb{R}^{2 \times 2}$ is an affine matrix. $\mathbf{w}_i$ is the deformation weight for the $i$-th TPS kernel function, $U_i(\cdot)$, which is defined as:

$$U(r) = r^2 \log r^2, \quad r = \|\mathbf{P}_x - \mathbf{P}_{s,i}\|_2. \tag{5}$$

Here, $r$ is the Euclidean distance between the feature point $\mathbf{p}_x$ and the $i$-th source control point.

The patch feature after deformation by $T(\cdot)$ is denoted as $\mathbf{F}_{\text{ltps}}$. We apply residual fusion with the original patch feature $F$:

$$\mathbf{F}_{\text{final}} = \mathbf{F} + \eta \cdot \mathbf{F}_{\text{ltps}}. \tag{6}$$

Here, $\eta$ is a tunable fusion factors. This fusion strategy allows the model to retain original semantic features while explicitly incorporating dynamic spatial structure awareness. As a result, it improves the model's ability to recognize people under occlusion, rotation, and deformation in complex scenes.

### 3.3 Dynamic Alignment Module

In person re-identification, training images often contain occlusions, background noise, or identity-irrelevant regions. Directly encoding features from the entire image may lead the model to learn redundant or non-discriminative representations. To address this issue, we explore whether the visibility patterns of a given input image can be used to dynamically modulate the feature representations of other samples. Inspired by DPM [40], we propose the **Dynamic Alignment Module (DAM)**, which leverages input-guided masks to improve semantic alignment. Specifically, DAM treats the mean feature representation of images with the same identity as its ideal prototype, assuming that noise or occlusion in different samples manifests as partial loss of this prototype's semantic structure. Based on this formulation, DAM introduces a dynamic channel-wise mask generator based on the input and modulates prototypes by referencing the input-aware mask. This allows the model to suppress irrelevant or noisy dimensions and emphasize input-relevant subspaces in a content-aware manner.

Specifically, during training, for each batch, we firstly construct prototype features for all classes as:

$$\mathbf{p}_c = \frac{1}{N_c} \sum_{i=1}^{N_c} \mathbf{f}_i. \tag{7}$$

Here, $N_c$ is the number of samples belonging to class $c$ within the current mini-batch, and $\mathbf{f}_i$ represents the feature embedding of the $i$-th sample. All prototype features are $\ell_2$-normalized both before and after the update to ensure consistent alignment with the sample features.

To enable dynamic selection of relevant prototype subspace, we design a channel-wise mask generator. This generator guides the model to identify and emphasize the prototype subspace most relevant for discriminating the current input image. Unlike spatial domain masks operating on pixel regions, we focus on channel-wise selection. The goal is to generate a sparse, discriminative channel-wise mask for each image, highlighting the discriminative dimensions during prototype matching.

The mask generator creates a channel-wise mask based on the current sample feature $\mathbf{f}_i$ with a two-layer multilayer perceptron (MLP), followed by a Sigmoid activation function to ensure mask values are between 0 and 1:

$$\mathbf{m}_i = \text{Sigmoid}\left(W_2 \cdot \text{ReLU}\left(W_1 \cdot \mathbf{f}_i + \mathbf{b_1}\right) + \mathbf{b_2}\right), \tag{8}$$

where $W_1 \in \mathbb{R}^{D \times \frac{D}{2}}$ and $W_2 \in \mathbb{R}^{\frac{D}{2} \times D}$ are learnable weight matrices, $\mathbf{b_1} \in \mathbb{R}^{\frac{D}{2}}$ and $\mathbf{b_2} \in \mathbb{R}^D$ denote the bias terms, and $\mathbf{f}_i$ is the input feature vector.

The generated sample-specific mask is then used to weight the prototype features, focusing on the parts of the features that are relevant to the category. The mask is applied by performing element-wise multiplication with the prototype features as follow:

$$\mathbf{p}_c^{masked} = \mathbf{p}_c \odot \mathbf{m}_c, \tag{9}$$

where $\mathbf{p}_c$ denotes the prototype feature of class $c$, and $\mathbf{m}_c$ is a channel-wise mask generated by the Channel-wise Mask Generator based on the feature representation of each sample and its corresponding label.

This mask is employed exclusively during the training phase to reweight the prototype $\mathbf{p}_c$ along the channel dimension. By emphasizing informative feature channels and suppressing less relevant ones, this mechanism encourages the model to focus on more discriminative aspects of the prototype representation. It is important to emphasize that this masked prototype weighting strategy is only applied during training to facilitate effective prototype learning. During inference, the model directly compares the extracted features with the unmasked prototypes, without relying on any class labels or mask generation.

Overall, the dynamic weighting mechanism enables the model to adaptively adjust prototype representations according to the feature distribution of each class. This enhances the discriminative capacity of the learned representations by selectively highlighting important channels and filtering out irrelevant dimensions. In addition, this module is utilized only during the training phase, which will not add the inference computation cost.

### 3.4 Loss Function and Optimization

Our model is trained using a composite loss function designed to optimize identity discrimination, geometric stability, and feature-level alignment. In addition to the standard ID classification loss $\mathcal{L}_{\text{id}}$ and the Triplet loss $\mathcal{L}_{\text{tri}}$ commonly employed in ReID tasks, we introduce a deformation loss $\mathcal{L}_{\text{deform}}$ and the mask loss $\mathcal{L}_{\text{mask}}$.

**Deformation Loss.** To enhance the stability of the LTPS module's deformation modeling, we introduce a regularization term as the deformation loss $\mathcal{L}_{\text{deform}}$. This loss discourages the rotation prediction submodule from outputting excessively large rotation angles, thereby promoting geometrically plausible transformations. While the angle prediction mechanism intrinsically limits angles via a $\tanh$ activation and scaling, this loss provides an additional constraint. It is defined as the average absolute predicted rotation angle across all $L$ encoder layers where an LTPS module is integrated:

$$\mathcal{L}_{\text{deform}} = \frac{1}{L} \sum_{l=1}^{L} |\theta_l|, \tag{10}$$

where $\theta_l$ represents the predicted rotation angle at the $l$-th layer. By penalizing large rotation magnitudes, this loss helps maintain geometric consistency during the feature transformation process.

**Mask Loss.** To enhance the effectiveness of the Dynamic Alignment Module, we introduce a mask alignment loss to ensure consistency between the dynamically masked prototype and the input sample feature. we formulate a composite mask loss $\mathcal{L}_{\text{mask}}$. This loss comprises two components: an alignment term $\mathcal{L}_{\text{align}}$ and an entropy-based regularization term $\mathcal{L}_{\text{entropy}}$:

$$\mathcal{L}_{\text{mask}} = \mathcal{L}_{\text{align}} + \lambda \mathcal{L}_{\text{entropy}}. \tag{11}$$

$\mathcal{L}_{\text{align}}$ aims to ensure that an input sample's feature representation, when masked, aligns closely with its corresponding class prototype, also masked by the same sample-specific mask. This encourages consistency in the feature subspace highlighted by the dynamic mask, even in the presence of occlusions or pose variations which might otherwise lead to disparate semantic activations. Let $\mathbf{f}_i$ be the feature of $i$-th sample with class of $c$, $\mathbf{m}_i$ its generated channel-wise mask, and $\mathbf{p}_c$ the prototype for the class $c$. The alignment loss is computed as:

$$\mathcal{L}_{\text{align}} = \frac{1}{N} \sum_{i=1}^{N} \|\text{Norm}(\mathbf{m}_i \odot \mathbf{p}_c) - \text{Norm}(\mathbf{m}_i \odot \mathbf{f}_i)\|_2^2, \tag{12}$$

where $N$ is the batch size, and Norm means the $\ell_2$ normalization. This loss encourages intra-class compactness within the dynamically selected channel subspace. By applying the mask $\mathbf{m}_i$ to both the sample feature and its prototype, the model learns to emphasize discriminative channels corresponding to visible regions while down-weighting channels associated with occlusions or background clutter.

$\mathcal{L}_{\text{entropy}}$ is designed to promote channel selectivity in the generated masks. During training, the mask tends to degenerate into a fully activated state, *i.e.*, $\mathbf{m}_i \rightarrow \mathbf{1}$. This loss drives the mask vector towards

a binary distribution by maximizing the information entropy, thus retaining the most relevant channels for each sample. It is defined as the negative sum of binary entropies for each mask component:

$$\mathcal{L}_{\text{entropy}} = -\frac{\lambda}{ND} \sum_{i=1}^{N} \sum_{d=1}^{D} \left[ \mathbf{m}_i^{(d)} \log \mathbf{m}_i^{(d)} + (1 - \mathbf{m}_i^{(d)}) \log(1 - \mathbf{m}_i^{(d)}) \right]. \tag{13}$$

$\mathcal{L}_{\text{align}}$ and $\mathcal{L}_{\text{entropy}}$ work as complementary constraints. $\mathcal{L}_{\text{align}}$ guides the mask to select channels that maximize prototype similarity, while $\mathcal{L}_{\text{entropy}}$ encourages the mask to make more definitive selections. A balancing hyperparameter $\lambda$ is introduced to regulate the trade-off between the two objectives.

**Overall Loss.** The model is trained end-to-end by minimizing a comprehensive loss function that aggregates the aforementioned components:

$$\mathcal{L}_{\text{total}} = (\mathcal{L}_{\text{id}} + \mathcal{L}_{\text{tri}}) + \alpha \mathcal{L}_{\text{deform}} + \beta \mathcal{L}_{\text{mask}}, \tag{14}$$

where the $\alpha$ and $\beta$ is the hyper-parameters to adjust the weight of $\mathcal{L}_{\text{deform}}$ and $\mathcal{L}_{\text{mask}}$ respectively. This joint optimization framework encourages the learning of discriminative and robust representations that are resilient to spatial misalignments and feature irrelevant variations.

## 4 Experiments

### 4.1 Implementation

**Datasets and Evaluation Protocols.** We conduct experiments on the CARGO dataset [8], a large-scale benchmark specifically designed for aerial-ground person re-identification (AG-ReID). CARGO contains 108,563 images of 5,000 synthetic identities captured by 13 cameras (5 aerial and 8 ground), with diverse conditions including extreme viewpoint changes, resolution variations, illumination shifts, and occlusions. The dataset is constructed in a simulated urban environment using MakeHuman for identity modeling and Unity3D for camera deployment.

Following the standard protocol, we use 51,451 images with 2500 identities for training and 51,024 images with the remaining 2,500 identities for testing. The testing phase contains four evaluation protocols: *ALL*, ground to ground( $G \leftrightarrow G$), aerial to aerial ( $A \leftrightarrow A$), and aerial to ground ($A \leftrightarrow G$), where each protocol focuses on a specific matching scenario. To further verify the generalization ability of GSAlign in real-world scenarios, we also conduct comparative experiments on the AG-ReID [6] and AG-ReID v2 [16] datasets, which contain real aerial and ground imagery captured in outdoor environments. We use Cumulative Matching Characteristics (CMC) at Rank-1 accuracy, mean Average Precision (mAP), and mean Inverse Negative Penalty (mINP) [41] as evaluation metrics.

**Implementation Details.** All implementation settings strictly follow the protocol of VDT [8] for fair comparison. We adopt ViT-Base as the backbone, initialized with ImageNet-21K pretraining. Input images are resized to $256 \times 128$, and horizontal flipping is applied during training for data augmentation. The model is optimized using AdamW with weight decay of 0.05 and a cosine learning rate schedule. The initial learning rate is set to $3.5 \times 10^{-4}$ with linear warm-up for the first 20 epochs. $\lambda$ is set to 0.1 during the training. We train for 120 epochs in total, using a batch size of 64.

### 4.2 Comparison with State-of-the-Arts

In this section, we compare our proposed **GSAlign** with a range of state-of-the-art person ReID methods, including both conventional CNN-based models (e.g., PCB [17], BoT [43], MGN [44]) and Transformer-based frameworks (e.g., ViT [39], VDT [8]). As shown in Table 1, GSAlign achieves the best overall performance across all four protocols. Specifically, under the most comprehensive *ALL* protocol, GSAlign achieves 65.06% Rank-1, 57.95% mAP, and 44.97% mINP, outperforming the strong baseline VDT [8] by +0.96%, +2.75%, and +3.84%, respectively. On the most challenging $A \leftrightarrow G$ cross-view setting, our method surpasses all previous methods by a large margin, reaching 64.89% Rank-1, 61.55% mAP, and 52.81% mINP, significantly ahead of the second-best VDT by +16.77%, +18.79%, and +22.86%, respectively. Although GSAlign is not specifically designed for single-view matching protocols, it does not introduce any negative effects in such settings. On both $A \leftrightarrow A$ and $G \leftrightarrow G$, GSAlign still achieves competitive performance. We attribute this to the fact that, in single-view scenarios, the features are already well aligned due to the limited viewpoint variation,

Table 1: **Performance comparison of the mainstream methods under four settings of the CARGO dataset**. "ALL" denotes the overall retrieval performance of each method. "G↔G", "A↔A", and "A↔G" represent the performance of each model in several specific retrieval patterns. Rank1, mAP, and mINP are reported (%). The best performance is shown in **bold**.

| Method | Protocol 1: ALL | | | Protocol 2: G↔G | | | Protocol 3: A↔A | | | Protocol 4: A↔G | | |
|---|---|---|---|---|---|---|---|---|---|---|---|---|
| | Rank1 | mAP | mINP | Rank1 | mAP | mINP | Rank1 | mAP | mINP | Rank1 | mAP | mINP |
| SBS [42] | 50.32 | 43.09 | 29.76 | 72.31 | 62.99 | 48.24 | 67.50 | 49.73 | 29.32 | 31.25 | 29.00 | 18.71 |
| PCB [17] | 51.00 | 44.50 | 32.20 | 74.10 | 67.60 | 55.10 | 55.00 | 44.60 | 27.00 | 34.40 | 30.40 | 20.10 |
| BoT [43] | 54.81 | 46.49 | 32.40 | 77.68 | 66.47 | 51.34 | 65.00 | 49.79 | 29.82 | 36.25 | 32.56 | 21.46 |
| MGN [44] | 54.81 | 49.08 | 36.52 | **83.93** | 71.05 | 55.20 | 65.00 | 52.96 | 36.78 | 31.87 | 33.47 | 24.64 |
| VV [45, 46] | 45.83 | 38.84 | 39.57 | 72.31 | 62.99 | 48.24 | 67.50 | 49.73 | 29.32 | 31.25 | 29.00 | 18.71 |
| AGW [41] | 60.26 | 53.44 | 40.22 | 81.25 | 71.66 | 58.09 | 67.50 | 56.48 | 40.40 | 43.57 | 40.90 | 29.39 |
| BAU [47] | 45.20 | 38.40 | - | 61.60 | 51.20 | - | 50.00 | 42.60 | - | 40.40 | 36.70 | - |
| PAT [48] | 37.90 | 15.30 | - | 52.70 | 24.20 | - | 50.00 | 23.10 | - | 35.10 | 15.50 | - |
| DTST [49] | 64.42 | 55.73 | 41.92 | 78.57 | 72.40 | 62.10 | 80.00 | 63.31 | 44.67 | 50.53 | 43.49 | 29.46 |
| ViT [39] | 61.54 | 53.54 | 39.62 | 82.14 | 71.34 | 57.55 | 80.00 | 64.47 | 47.07 | 43.13 | 40.11 | 28.20 |
| VDT [8] | 64.10 | 55.20 | 41.13 | 82.14 | 71.59 | 58.39 | **82.50** | **66.83** | **50.22** | 48.12 | 42.76 | 29.95 |
| GSAlign | **65.06** | **57.95** | **44.97** | 83.04 | **73.86** | **62.73** | 80.00 | 65.55 | 49.81 | **64.89** | **61.55** | **52.81** |

thus requiring minimal additional geometric or semantic correction. In conclusion, these results clearly demonstrate that our proposed GSAlign not only achieves strong global retrieval accuracy but also exhibits exceptional robustness in the presence of extreme viewpoint changes, validating the effectiveness of our geometric and semantic alignment design.

Furthermore, we evaluate the proposed GSAlign on the real-world AG-ReID [6] and AG-ReID v2 [16] datasets, and compare it with a wide range of state-of-the-art ReID methods, as summarized in Table 2 and Table 3. On the AG-ReID dataset, GSAlign achieves strong and consistent performance under both cross-view protocols. Under the A↔G setting, GSAlign obtains the best results across all metrics, reaching 83.75% Rank-1, 75.01% mAP, and 52.11% mINP, outperforming the view-aware baseline VDT by +0.84%, +0.57%, and +1.05%, respectively. Under the inverse G↔A protocol, GSAlign achieves competitive Rank-1 (84.10%) and mAP (77.73%) scores, while yielding the highest mINP of 53.63%, demonstrating its effec-

Table 2: **Performance comparison of the mainstream methods under two settings of the AG-ReID dataset.** Results are reported under four matching protocols. The best values per column are shown in **bold**.

| Setting | Protocol 1: A↔G | | | Protocol 2: G↔A | | |
|---|---|---|---|---|---|---|
| | Rank1 | mAP | mINP | Rank1 | mAP | mINP |
| SBS [42] | 73.54 | 59.77 | - | 73.70 | 62.27 | - |
| BoT [43] | 70.01 | 55.47 | - | 71.20 | 58.83 | - |
| OsNet [50] | 72.59 | 58.32 | - | 74.22 | 60.99 | - |
| VV [45, 46] | 77.22 | 67.23 | 41.43 | 79.73 | 69.83 | 42.37 |
| Explain | 81.28 | 72.38 | - | 82.64 | 73.35 | - |
| ViT [39] | 81.47 | 72.61 | - | 82.85 | 73.39 | - |
| VDT [8] | 82.91 | 74.44 | 51.06 | **86.59** | **78.57** | 52.87 |
| **GSAlign** | **83.75** | **75.01** | **52.11** | 84.10 | 77.73 | **53.63** |

tiveness in hard-sample and long-tail retrieval under severe viewpoint changes. On the more challenging AG-ReID v2 dataset, which features more realistic and diverse scenes, GSAlign continues to exhibit strong generalization ability. As shown in Table 3, GSAlign achieves the highest mAP across all four protocols, including A↔G, G↔A, A↔W, and W↔A, with mAP values of 81.38%, 81.05%, 83.98%, and 80.90%, respectively. Although some methods attain slightly higher Rank-1 accuracy under specific settings, GSAlign remains highly competitive while delivering more stable overall retrieval performance.

In summary, the experimental results on both AG-ReID and AG-ReID v2 datasets demonstrate that GSAlign effectively enhances cross-view feature alignment and retrieval robustness. The consistent gains in mAP and mINP across diverse protocols validate the effectiveness and robustness of our geometric and semantic alignment design in real-world aerial–ground ReID scenarios.

## 4.3 Ablation Study

To assess the individual contributions of each component in the proposed GSAlign framework, we conduct a comprehensive ablation study on the four testing protocols of the CARGO dataset. As shown in Table 4, we start with a strong transformer-based baseline [8] and progressively integrate

Table 3: **Performance comparison of the mainstream methods under four settings of the AG-ReID v2 dataset.** Results are reported under four matching protocols. The best values per column are shown in **bold**.

| Setting | Protocol 1: A↔G | | Protocol 2: G↔A | | Protocol 3: A↔W | | Protocol 4: W↔A | |
|---|---|---|---|---|---|---|---|---|
| | Rank1 | mAP | Rank1 | mAP | Rank1 | mAP | Rank1 | mAP |
| BoT [43] | 85.40 | 77.03 | 84.65 | 75.90 | 89.77 | 80.48 | 84.65 | 76.90 |
| Explain [6] | 87.70 | 79.00 | 87.35 | 78.24 | 93.67 | 83.14 | 87.73 | 79.08 |
| AG-ReIDv2 [16] | **88.77** | 80.72 | 87.86 | 78.51 | **93.62** | **84.85** | **88.61** | 80.11 |
| SeCap [51] | 88.12 | 80.84 | **88.24** | 79.99 | 91.44 | 84.01 | 87.56 | 80.15 |
| VDT [8] | 86.46 | 79.13 | 86.14 | 78.12 | 90.00 | 82.21 | 85.26 | 78.52 |
| **GSAlign** | 87.86 | **81.38** | 88.02 | **81.05** | 90.63 | 83.98 | 87.31 | **80.90** |

Table 4: **Ablation study of the different components in GSAlign on the CARGO dataset.** "LTPS" denotes the Learnable Thin Plate Spline transformation module. "DAM" refers to the Dynamic Alignment Module. The best performance per column is shown in **bold**.

| Setting | Protocol 1: ALL | | | Protocol 2: G↔G | | | Protocol 3: A↔A | | | Protocol 4: A↔G | | |
|---|---|---|---|---|---|---|---|---|---|---|---|---|
| | Rank1 | mAP | mINP | Rank1 | mAP | mINP | Rank1 | mAP | mINP | Rank1 | mAP | mINP |
| Baseline | 64.10 | 55.20 | 41.13 | 82.14 | 71.59 | 58.39 | **82.50** | **66.83** | **50.22** | 48.12 | 42.76 | 29.95 |
| Baseline + LTPS | 64.42 | 55.95 | 41.92 | 80.36 | 71.87 | 59.55 | **82.50** | 65.26 | 47.15 | **64.89** | 61.08 | 50.54 |
| Baseline + LTPS + DAM | **65.06** | **57.95** | **44.97** | **83.04** | **73.86** | **62.73** | 80.00 | 65.55 | 49.81 | **64.89** | **61.55** | **52.81** |

the Learnable Thin Plate Spline (LTPS) transformation module and the Dynamic Alignment Module (DAM). Compared to the baseline, integrating LTPS brings consistent performance improvements across most settings. Specifically, on the most challenging A↔G protocol, LTPS improves Rank-1 from 48.12% to 64.89%, mAP from 42.76% to 61.08%, and mINP from 29.95% to 50.54%. These results confirm the effectiveness of progressive geometric alignment in compensating for severe cross-view deformations. Interestingly, the performance gain on A↔A and G↔G is more modest, indicating that LTPS is particularly beneficial when strong viewpoint discrepancies exist. Adding the DAM module on top of LTPS yields further improvements, especially in terms of mAP and mINP. Under the full matching scenario (Protocol 1: ALL), the full model achieves the best performance with 65.06% Rank-1, 57.95% mAP, and 44.97% mINP. Notably, the largest relative gain comes from the A↔G setting, where DAM increases mINP from 50.54% to 52.81%. This demonstrates that DAM effectively suppresses noisy and occluded regions by leveraging visibility-aware semantic masking, thus enhancing cross-view alignment. Overall, the ablation study verifies the complementary effects of LTPS and DAM. While LTPS resolves geometric distortions in a layer-wise manner, DAM provides semantic-level refinement by selectively filtering out unreliable features. Their combination allows GSAlign to maintain robust performance across diverse matching protocols, especially under severe viewpoint and occlusion conditions.

## 4.4 Discussions[3]

**Number of control points in LTPS.** To examine the sensitivity of GSAlign to the number of control points in the Learnable Thin Plate Spline (LTPS) module, we test five settings with 4, 9, 16, 25, 36 keypoints on the CARGO dataset under four protocols. As shown in Table 5, all variants benefit from LTPS-based geometric alignment, though performance varies with point density. The 4-point LTPS achieves the best overall results as Rank-1 (64.89%), mAP (61.55%), and mINP (52.81%) under the A↔G protocol and remains strong across all scenarios. Increasing points (e.g., 25 or 36) slightly improves Rank-1 but reduces mAP and mINP, likely due to over-flexibility and local distortion. These findings indicate that a lightweight 4-point LTPS provides an optimal trade-off between alignment accuracy and stability.

**Different locations for LTPS.** We further study the effect of inserting LTPS at different depths of the ViT backbone. As shown in Table 6, applying LTPS to all transformer layers yields the best results (64.89% Rank-1, 61.55% mAP, 52.81% mINP under Protocol 4: A↔G) and consistent gains across scenarios. Among partial insertions, placing LTPS in the last 4 layers performs comparably to the full setting, implying that geometric distortions persist even in higher-level features and benefit

---

[3]A visualization results of LTPS (Sec. A.1) is provided in the supplementary material.

Table 5: **Performance of GSAlign under different numbers of control points in LTPS.** Results are reported under four matching protocols. The best values per column are shown in **bold**.

| Setting | Protocol 1: ALL | | | Protocol 2: G↔G | | | Protocol 3: A↔A | | | Protocol 4: A↔G | | |
|---|---|---|---|---|---|---|---|---|---|---|---|---|
| | Rank1 | mAP | mINP | Rank1 | mAP | mINP | Rank1 | mAP | mINP | Rank1 | mAP | mINP |
| **Number of control points in LTPS** | | | | | | | | | | | | |
| 4 | 65.06 | **57.95** | **44.97** | **83.04** | 73.86 | 62.73 | 80.00 | 65.55 | **49.81** | **64.89** | **61.55** | **52.81** |
| 9 | 64.10 | 57.35 | 44.76 | 82.14 | **75.16** | **64.91** | 80.00 | 64.51 | 47.23 | 62.77 | 59.25 | 50.19 |
| 16 | 63.78 | 56.48 | 42.97 | 80.36 | 73.78 | 63.49 | 77.50 | 63.48 | 47.28 | 61.70 | 58.06 | 48.19 |
| 25 | **65.71** | 57.62 | 44.15 | 82.14 | 74.06 | 62.93 | **82.50** | **65.95** | 48.20 | 63.83 | 60.23 | 50.53 |
| 36 | **65.71** | 57.55 | 44.34 | 81.25 | 72.71 | 61.11 | **82.50** | 65.88 | 48.47 | 63.83 | 60.00 | 51.12 |

Table 6: **Comparison between different locations for LTPS.** Results are reported under four matching protocols. The best values per column are shown in **bold**.

| Setting | Protocol 1: ALL | | | Protocol 2: G↔G | | | Protocol 3: A↔A | | | Protocol 4: A↔G | | |
|---|---|---|---|---|---|---|---|---|---|---|---|---|
| | Rank1 | mAP | mINP | Rank1 | mAP | mINP | Rank1 | mAP | mINP | Rank1 | mAP | mINP |
| **Different locations for LTPS** | | | | | | | | | | | | |
| First layer | 64.10 | 55.92 | 42.44 | **83.04** | 72.86 | 60.58 | **80.00** | **65.98** | **50.45** | 58.51 | 56.92 | 47.62 |
| First 4 layers | 64.10 | 56.46 | 43.50 | 81.25 | 74.49 | 64.70 | **80.00** | 64.45 | 47.11 | 58.51 | 56.21 | 46.66 |
| Middle 4 layers | 64.74 | 57.09 | 44.08 | 82.14 | **74.93** | **64.77** | 77.50 | 64.28 | 47.30 | 58.51 | 58.30 | 50.18 |
| Last 4 layers | **65.06** | 57.39 | 44.05 | **83.04** | 74.42 | 62.86 | 77.50 | 65.21 | 49.80 | **64.89** | 59.87 | 50.95 |
| All layers | **65.06** | **57.95** | **44.97** | **83.04** | 73.86 | 62.73 | **80.00** | 65.55 | 49.81 | **64.89** | **61.55** | **52.81** |

Table 7: **Comparison between different variants of DAM.** Results are reported under four matching protocols. The best values per column are shown in **bold**.

| Setting | Protocol 1: ALL | | | Protocol 2: G↔G | | | Protocol 3: A↔A | | | Protocol 4: A↔G | | |
|---|---|---|---|---|---|---|---|---|---|---|---|---|
| | Rank1 | mAP | mINP | Rank1 | mAP | mINP | Rank1 | mAP | mINP | Rank1 | mAP | mINP |
| **Different variants of DAM** | | | | | | | | | | | | |
| Inner-Batch | 65.06 | **57.95** | **44.97** | **83.04** | 73.86 | **62.73** | **80.00** | **65.55** | **49.81** | **64.89** | **61.55** | **52.81** |
| Memory Bank | **65.38** | 57.34 | 44.09 | **83.04** | 73.72 | 62.05 | **80.00** | 62.70 | 43.88 | 63.83 | 61.06 | 52.52 |
| Classification Matrix | 63.14 | 55.64 | 42.07 | 81.25 | 72.04 | 59.53 | 75.00 | 63.79 | 48.06 | 57.45 | 56.55 | 47.33 |

from late-stage correction. In contrast, early-layer insertion offers limited improvement, especially in fine-grained metrics like mINP, likely because shallow features are less discriminative and harder to align precisely. These results highlight the importance of progressive alignment from shallow to deep layers for handling complex viewpoint distortions in AG-ReID.

**Different variants of DAM.** We compare three variants of the Dynamic Alignment Module (DAM): Inner-Batch, Memory Bank, and Classification Matrix. As shown in Table 7, the Inner-Batch variant achieves the best overall performance. It dynamically generates visibility-aware masks within each mini-batch, enabling query-driven refinement of peer features. The Memory Bank variant stores identity prototypes updated by momentum, providing semantic context but suffering from stale or mismatched memory. The Classification Matrix variant reuses classifier weights as class centers [40], but these prototypes are biased toward classification objectives. Overall, Inner-Batch masking proves most effective for dynamic alignment in aerial-ground person ReID.

# 5   Conclusion

In this paper, we propose GSAlign, a novel framework for aerial-ground person re-identification that jointly addresses geometric distortion and semantic misalignment. By introducing a Learnable Thin Plate Spline (LTPS) module and a Dynamic Alignment Module (DAM), GSAlign performs progressive geometric correction and visibility-aware semantic filtering in a unified manner. Extensive experiments on the challenging CARGO and AG-ReID datasets demonstrate that GSAlign achieves state-of-the-art performance across multiple protocols, highlighting its effectiveness in handling severe viewpoint changes and partial occlusions.

**Acknowledgements.** This research was supported in part by the National Key R&D Program of China under grant No. 2022YFB3103300.

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

# A  Supplementary Material

## A.1  Visualization results of LTPS

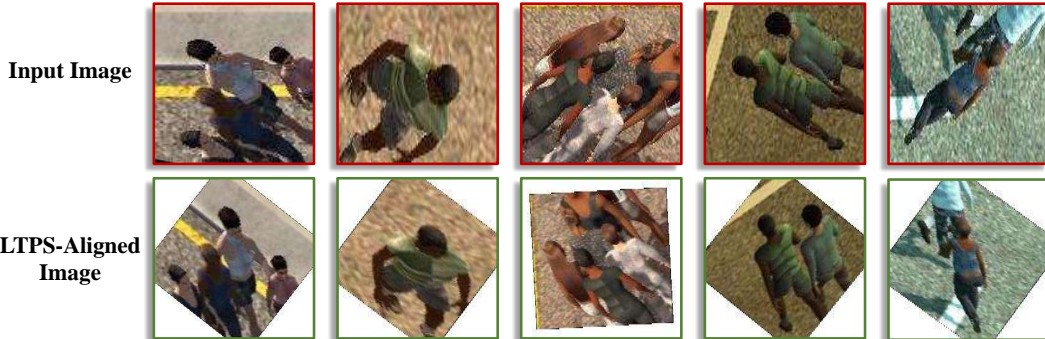

Figure 3: **Qualitative comparison before and after LTPS alignment.** The input image (red) exhibits significant geometric distortion due to extreme viewpoint variation. After applying the Learnable Thin Plate Spline (LTPS) transformation (green), the image is spatially rectified, highlighting improved geometric consistency and local structure alignment.

To further demonstrate the effectiveness of our LTPS strategy, we visualize LTPS outputs in Fig. 3. To facilitate better visualization, we aggregate the LTPS transformations from all layers and apply the fused transformation to the original input image. As shown in Fig. 3., the input images exhibit severe geometric distortions caused by extreme aerial-ground viewpoint differences. After applying the Learnable Thin Plate Spline (LTPS) transformation, the visual structure of the person becomes significantly more regular and consistent. These rectified features allow the network to focus on semantically consistent regions and reduce the burden of learning viewpoint-invariant representations purely through data. This visual evidence supports the quantitative gains observed in our ablation studies and highlights the role of LTPS in improving cross-view structural correspondence.

## A.2  Effectiveness Analysis of the LTPS Module

**Effect of the Deformation Constraint**. We evaluate the role of the deformation constraint $L_{\text{deform}}$ in our LTPS module. As shown in Table 8, we consider three variants: (1) Fixed-angle rotation, where a constant rotation is applied without learnable deformation; (2) LTPS without $L_{\text{deform}}$, where the rotation angle is freely optimized under the supervision of the ReID loss alone; and (3) LTPS with $L_{\text{deform}}$, our final proposed design that jointly optimizes the rotation angle under both the ReID loss and the deformation constraint. The comparison reveals that unconstrained optimization of rotation angles can easily lead to excessive or unstable transformations, degrading feature alignment and recognition performance. In contrast, incorporating $L_{\text{deform}}$ effectively regularizes the deformation process, preventing overfitting to local minima and improving global spatial consistency. As a result, LTPS with $L_{\text{deform}}$ achieves the best performance across all CARGO evaluation protocols, demonstrating the necessity of this constraint in maintaining geometric stability during feature learning.

Table 8: **Performance comparison of fixed-angle rotation and LTPS on the CARGO dataset.** Results are reported under four matching protocols. The best values per column are shown in **bold**.

| Setting | Protocol 1: ALL | | | Protocol 2: G↔G | | | Protocol 3: A↔A | | | Protocol 4: A↔G | | |
|---|---|---|---|---|---|---|---|---|---|---|---|---|
| | Rank1 | mAP | mINP | Rank1 | mAP | mINP | Rank1 | mAP | mINP | Rank1 | mAP | mINP |
| **Different rotation of GSAlign** | | | | | | | | | | | | |
| fixed-angle rotation | 60.65 | 52.06 | 38.83 | 78.89 | 68.21 | 57.64 | 77.50 | 59.54 | 40.73 | 58.45 | 53.74 | 43.84 |
| LTPS without $\mathcal{L}_{\text{deform}}$ | 59.94 | 52.67 | 39.25 | 76.79 | 69.83 | 57.68 | 75.00 | 43.49 | 43.49 | 56.38 | 53.22 | 43.96 |
| LTPS with $\mathcal{L}_{\text{deform}}$ | **66.67** | **56.35** | **41.42** | **83.04** | **71.57** | **57.68** | **82.50** | **67.70** | **51.80** | **65.96** | **60.60** | **49.32** |

**Comparison with Original TPS.** We further compare the proposed LTPS with the original TPS, which applies standard TPS transformations at every layer. The experiments are conducted on the CARGO dataset, and the results are summarized in Table 9. The results indicate that while the original TPS can partially mitigate feature misalignment in the $A \leftrightarrow G$ setting, its fixed rotation and interpolation strategy fail to generalize well to the $G \leftrightarrow G$ scenario. For example, although GSAlign with original TPS achieves minor gains under $A \rightarrow G$, it suffers from noticeable degradation under $G \leftrightarrow G$, leading to an overall performance even lower than that of ViT. In contrast, our LTPS incorporates learnable control points and a deformation regularization term that

constrains rotation magnitudes, effectively preventing excessive geometric distortion. This design allows LTPS to maintain strong adaptability under $A \leftrightarrow G$ conditions while significantly enhancing retrieval accuracy under $G \leftrightarrow G$ scenarios. Overall, LTPS achieves more stable and balanced performance across different view settings, making it a more effective and generalizable solution than the standard TPS approach.

Table 9: **Performance comparison of original TPS and LTPS on CARGO dataset.** Here, GSAlign-O is ViT applying original TPS transformations at every layer.

| Setting | Protocol 1: ALL | | | Protocol 2: G↔G | | | Protocol 3: A↔A | | | Protocol 4: A↔G | | |
|---|---|---|---|---|---|---|---|---|---|---|---|---|
| | Rank1 | mAP | mINP | Rank1 | mAP | mINP | Rank1 | mAP | mINP | Rank1 | mAP | mINP |
| GSAlign-O | 63.78 | 55.18 | 41.63 | 80.36 | 71.21 | 57.06 | 82.50 | 65.44 | 47.37 | 58.51 | 57.72 | 49.02 |
| GSAlign | 66.67 | 56.35 | 41.42 | 83.04 | 71.57 | 58.68 | 82.50 | 67.70 | 51.80 | 65.96 | 60.50 | 49.32 |

**Efficiency Analysis of LTPS.** To assess the computational efficiency of the proposed LTPS module, we compare the inference cost between the baseline ViT model and the LTPS-integrated GSAlign. The results show that both models have identical FLOPs of 17.67 GFLOPs, indicating that the additional computational cost introduced by LTPS is negligible. This is mainly because the Thin Plate Spline (TPS) transformation in LTPS is lightweight—its computation is minimal compared to the large-scale matrix multiplications in the ViT backbone. Consequently, the LTPS-enhanced GSAlign maintains nearly the same theoretical inference cost as the original ViT, ensuring real-time efficiency. We further measure the practical inference speed under identical experimental conditions. Using a batch size of 128, GSAlign achieves an average inference time of 0.791 seconds per batch, while the baseline ViT-based model requires 0.778 seconds per batch. The minor difference of 0.013 seconds demonstrates that integrating LTPS at each layer introduces no significant runtime overhead, confirming the model's suitability for real-time deployment.

## A.3 Limitation and Boroader Impact

Despite the promising results of the proposed Dynamic Cross-view Alignment framework in aerial–ground person re-identification, there are still some limitations. First, the performance of the LTPS module depends on the accuracy of predicted control point offsets and rotation angles. Under extreme viewpoint changes or heavy occlusions, the model may fail to learn reliable spatial transformations, which can affect feature alignment. Moreover, in aerial–aerial scenarios where geometric variations are relatively small and pedestrians appear with limited texture information, the LTPS and DAM modules may introduce unnecessary local transformations, leading to a slight performance drop. This is because these modules are primarily designed to handle large spatial misalignments in the aerial–ground setting. In addition, our method is currently evaluated on specific aerial–ground datasets; however, experiments on real-world scenarios demonstrate that the framework remains effective beyond these datasets. Nevertheless, we believe the proposed framework provides a promising direction for modeling spatial variations and focusing on identity-relevant regions in cross-view re-identification.

