# OpenReview forum: "GSAlign: Geometric and Semantic Alignment Network for Aerial-Ground Person Re-Identification"
_NeurIPS.cc/2025/Conference — NeurIPS 2025 poster_

### Official Review · Reviewer_GGnw · 2025-06-23

**Clarity:** 3
**Significance:** 2
**Originality:** 3
**Rating:** 4
**Confidence:** 4

**Summary:**

This paper proposes GSAlign, a person re-identification (ReID) framework specifically designed for synthesized datasets featuring extreme viewpoint variations and occlusions, such as those from UAV footage or virtual environments. Experiments on the synthetic CARGO dataset show that GSAlign achieves good performance.

**Questions:**

1. Have you tried GSAlign on real-world datasets? If not, what are your expectations regarding generalization beyond synthetic domains?
2. What is the additional inference cost (such as FLOPs) introduced by LTPS per layer?

**Ethical Concerns:**

["NO or VERY MINOR ethics concerns only"]

**Final Justification:**

Thanks for authors’ response and I will maintain my orignal rating score.

**Limitations:**

As mentioned in the Cons: The framework is validated only on synthetic data, limiting its practical impact unless real-world evaluations are added. This method implicitly depends on accurate keypoint detection, which can be unreliable under occlusions or low resolution.

**Quality:**

3

**Strengths And Weaknesses:**

Strength:
The paper is well-motivated. It tackles a specific and practically important problem: ReID under extreme camera angles and occlusions by leveraging synthetic data. The strong results on CARGO dataset is quite confident.

Weaknesses:
1. Experiments are conducted only on the synthetic CARGO dataset. Despite its diversity, there is no validation on real-world benchmarks, making it unclear whether the method generalizes beyond virtual domains.
2. DAM is used solely during training, while inference uses the unmasked prototypes. The authors do not evaluate whether DAM improves inference features or whether it merely functions as a training regularizer.
3. The paper lacks an analysis of the additional computational cost of applying LTPS at every ViT layer. This may hinder real-time or edge deployment.
4. The model implicitly depends on accurate keypoint detection, which can be unreliable under occlusions or low resolution scenarios, which are common in real-world UAV data.
Despite the weaknesses, I am looking forward to the authors’ reply.

---

> ### Author Rebuttal · Authors · 2025-07-31
>
> Thanks for your constructive feedback. Below, we respond to your key concerns point by point.
> > Q1:  Have you tried GSAlign on real-world datasets? If not, what are your expectations regarding generalization beyond synthetic domains?
> >
>
> A1: Following your suggestion, we conducted comprehensive comparative experiments on the real-world AG-ReID dataset [1], covering both cross-domain settings: A→G and G→A. As shown in Table I, extensive experiments on this challenging real-world dataset demonstrate the effectiveness of GSAlign. Compared to previous state-of-the-art methods, our proposed GSAlign still achieves significant improvements.
>
> Table I: Performance comparison of the mainstream methods under two settings of the AG-ReID dataset. Here, VDT-R is the reproduced version of its open-source code.
>
> | Method | A→G |  |  | G→A |  |  |
> | --- | --- | --- | --- | --- | --- | --- |
> |  | Rank1 | mAP | mINP | Rank1 | mAP | mINP |
> | SBS [2] | 73.54 | 59.77 | - | 73.70 | 62.27 | - |
> | BoT [3] | 70.01 | 55.47 | - | 71.20 | 58.83 | - |
> | OsNet [4] | 72.59 | 58.32 | - | 74.22 | 60.99 | - |
> | VV [5] | 77.22 | 67.23 | 41.43 | 79.73 | 69.83 | 42.37 |
> | AG-ReIDv1 [1] | 81.28 | 72.38 | - | 82.64 | 73.35 | - |
> | ViT [6] | 81.47 | 72.61 | - | 82/85 | 73.39 | - |
> | VDT [7] | 82.91 | 74.44 | 51.06 | 86.59 | 78.57 | 52.87 |
> | VDT-R [7] | 82.73 | 74.30 | 51.77 | 84.62 | 76.56 | 49.70 |
> | GSAlign | **86.74** | **84.00** | **73.62** | **87.94** | **87.17** | **78.21** |
>
> [1]Nguyen, Huy, et al. "Aerial-ground person re-id." ICME, 2023.
>
> [2]He, Lingxiao, et al. "Fastreid: A pytorch toolbox for general instance re-identification." MM, 2023.
>
> [3]Luo, Hao, et al. "Bag of tricks and a strong baseline for deep person re-identification." CVPR, 2019.
>
> [4]Zhou, Kaiyang, et al. "Learning generalisable omni-scale representations for person re-identification." TPAMI, 2021.
>
> [5]Kuma, Ratnesh, et al. "Vehicle re-identification: an efficient baseline using triplet embedding." IJCNN, 2019.
>
> [6]Dosovitskiy, Alexey, et al. "An image is worth 16x16 words: Transformers for image recognition at scale." ICLR, 2021
>
> [7]Zhang, Quan, et al. "View-decoupled transformer for person re-identification under aerial-ground camera network." CVPR, 2024.
>
> > Q2:  What is the additional inference cost (such as FLOPs) introduced by LTPS per layer?
> >
>
> A2:  We used the fvcore library to compare the FLOPs of the original ViT and our GSAlign model with the LTPS module integrated. **The results show that both models have identical FLOPs of 17.67 GFLOPs, indicating that the additional computational overhead introduced by LTPS is negligible.** This outcome is attributed to the lightweight nature of the Thin Plate Spline (TPS) transformation used in LTPS. Compared to the large-scale matrix multiplications in the ViT backbone, the computation required by TPS is minimal and has little impact on the overall FLOPs. Therefore, the LTPS-enhanced model maintains nearly the same inference cost as the original ViT, ensuring real-time performance. Besides, we conducted an inference speed comparison between the baseline VDT model and our GSAlign model with LTPS under the same environment (Ubuntu 20.04, Intel Xeon Gold 6133 CPU @ 2.50GHz, NVIDIA RTX 4090 GPU, Python 3.8, PyTorch 1.12.0). **With a batch size of 128, GSAlign achieved an average inference time of 0.791 seconds per batch, while the baseline ViT-based model achieved 0.778 seconds per batch.** The difference in inference time is minimal, indicating that integrating LTPS at each layer does not introduce significant computational overhead, and the model retains strong potential for real-time deployment.
>
> > Q3: DAM is used solely during training, while inference uses the unmasked prototypes. The authors do not evaluate whether DAM improves inference features or whether it merely functions as a training regularizer.
> >
>
> A3: The primary purpose of the DAM module is to serve as a regularization mechanism during training, rather than as a feature generator during inference. During training, DAM applies dynamic masking constraints in the feature space, encouraging the model to learn more discriminative and generalizable representations, thereby improving overall performance. However, in the inference phase, it is not feasible to dynamically generate masks for the test set due to the absence of label information and the need to maintain efficient parallel computation. Therefore, we perform retrieval using the unmasked prototypes. As a result, the effect of DAM lies mainly in its regularization role during training, rather than altering the feature distribution at inference. Its benefit is reflected indirectly through overall performance gains, rather than through separate measurements on inference-time features.
>
> > Q4: The model implicitly depends on accurate keypoint detection, which can be unreliable under occlusions or low resolution scenarios, which are common in real-world UAV data. Despite the weaknesses, I am looking forward to the authors’ reply.
> >
>
> A4:  We thank the reviewer for the insightful question. Indeed, the model’s performance can be partially influenced by the accuracy of keypoint detection, which may be challenged by common issues in UAV-captured imagery such as occlusion and low resolution.
> However, our work primarily focuses on addressing the cross-view misalignment problem between aerial and ground perspectives in person re-identification. While keypoint accuracy is important, it is not the central objective of this study. Through our experiments, we demonstrate the effectiveness and robustness of our approach in mitigating viewpoint misalignment.
> To better handle low-quality images, we plan to explore the integration of super-resolution techniques, occlusion detection, and feature restoration in future work. These enhancements could improve image quality and model generalization, thereby boosting performance in challenging UAV-based scenarios. We look forward to advancing these directions in future research to further support real-world deployment.

---

> > ### Comment · Area_Chair_icEs · 2025-08-07
> > **Discussion**
> >
> > Dear Reviewer GGnw,
> >
> > Thank you for your service to NeruIPS 2025 paper review. What do you think of the author rebuttal? Did they address your concerns? Could you please kindly help share your further opinions? Thank you.
> >
> > Best regards,
> > Your AC

---

> > ### Comment · Reviewer_GGnw · 2025-08-08
> >
> > Thanks for your response. My concerns have been well addressed.

---

### Official Review · Reviewer_5Ks3 · 2025-06-27

**Clarity:** 3
**Significance:** 2
**Originality:** 4
**Rating:** 5
**Confidence:** 4

**Summary:**

This paper proposes GSAlign, a novel framework for aerial-ground person re-identification (AG-ReID), addressing the challenges of extreme viewpoint variations and occlusions. GSAlign introduces two key components: a learnable thin plate spline (LTPS) module and a dynamic alignment module (DAM). Built on a Vision Transformer backbone, the model adaptively corrects spatial distortions and highlights visible body regions. Experiments on the CARGO dataset show significant improvements over prior methods, particularly in challenging cross-view scenarios, demonstrating robustness and effectiveness of GSAlign in AG-ReID tasks.

**Questions:**

1. How does the proposed method perform on the AGReID dataset? AGReID is a widely used and real-world benchmark for Aerial-Ground Person Re-Identification, and evaluating on it would better validate the practical effectiveness of the approach.
2. Eq. (6) introduces a fusion factor. How does it work? Is it a learnable parameter or a handcraft parameter? If it is a handcraft parameter, how you get the value?
3. Compared to the original TPS, the proposed LTPS makes some rules during the training. So, how does this change influence the performance when compared to the original TPS?

[1] View-decoupled transformer for person re-identification under aerial-ground camera network. CVPR 2024.

**Ethical Concerns:**

["NO or VERY MINOR ethics concerns only"]

**Final Justification:**

The experiment on the AG-ReiD dataset and hyperparameters has addressed some of my concerns. I have decided to maintain this positive score.

**Limitations:**

Yes

**Quality:**

3

**Strengths And Weaknesses:**

Strengths:
1. This paper is well-organized from its motivation and method. The problem is specific with reasonable module design and strong performance improvement.
2. The idea of learnable thin plate spline (LTPS) is innovation on this topic. Visualization also shows interesting results.

Weaknesses:
1. In this paper, the authors evaluate GSAlign solely on the synthetic CARGO dataset. I suggest the authors include results on real-world datasets such as AG-ReID [1] to better demonstrate the method’s generalizability.
2. The paper lacks some details, such as the selection of hyperparameters and evidence of the effectiveness of the proposed improvements. While I believe these designs are theoretically sound, supporting experimental results are necessary to validate them.

[1] View-decoupled transformer for person re-identification under aerial-ground camera network. CVPR 2024.

---

> ### Author Rebuttal · Authors · 2025-07-31
>
> Thanks for your constructive feedback. Below, we respond to your key concerns point by point.
> > **Q1:**  How does the proposed method perform on the AGReID dataset? AGReID is a widely used and real-world benchmark for Aerial-Ground Person Re-Identification, and evaluating on it would better validate the practical effectiveness of the approach.
> >
>
> **A1:** Following your suggestion, we conducted comprehensive comparative experiments on the AG-ReID dataset [1], covering both cross-domain settings: A→G and G→A. As shown in Table F, extensive experiments on this challenging real-world dataset demonstrate the effectiveness of GSAlign. Compared to previous state-of-the-art methods, our proposed GSAlign still achieves significant improvements.
>
> Table F: Performance comparison of the mainstream methods under two settings of the AG-ReID dataset. Here, VDT-R is the reproduced version of its open-source code.
>
> | Method | A→G |  |  | G→A |  |  |
> | --- | --- | --- | --- | --- | --- | --- |
> |  | Rank1 | mAP | mINP | Rank1 | mAP | mINP |
> | SBS [2] | 73.54 | 59.77 | - | 73.70 | 62.27 | - |
> | BoT [3] | 70.01 | 55.47 | - | 71.20 | 58.83 | - |
> | OsNet [4] | 72.59 | 58.32 | - | 74.22 | 60.99 | - |
> | VV [5] | 77.22 | 67.23 | 41.43 | 79.73 | 69.83 | 42.37 |
> | AG-ReIDv1 [1] | 81.28 | 72.38 | - | 82.64 | 73.35 | - |
> | ViT [6] | 81.47 | 72.61 | - | 82/85 | 73.39 | - |
> | VDT [7] | 82.91 | 74.44 | 51.06 | 86.59 | 78.57 | 52.87 |
> | VDT-R [7] | 82.73 | 74.30 | 51.77 | 84.62 | 76.56 | 49.70 |
> | GSAlign | **86.74** | **84.00** | **73.62** | **87.94** | **87.17** | **78.21** |
>
> [1]Nguyen, Huy, et al. "Aerial-ground person re-id." ICME, 2023.
>
> [2]He, Lingxiao, et al. "Fastreid: A pytorch toolbox for general instance re-identification." MM, 2023.
>
> [3]Luo, Hao, et al. "Bag of tricks and a strong baseline for deep person re-identification." CVPR, 2019.
>
> [4]Zhou, Kaiyang, et al. "Learning generalisable omni-scale representations for person re-identification." TPAMI, 2021.
>
> [5]Kuma, Ratnesh, et al. "Vehicle re-identification: an efficient baseline using triplet embedding." IJCNN, 2019.
>
> [6]Dosovitskiy, Alexey, et al. "An image is worth 16x16 words: Transformers for image recognition at scale." ICLR, 2021
>
> [7]Zhang, Quan, et al. "View-decoupled transformer for person re-identification under aerial-ground camera network." CVPR, 2024.
>
> > **Q2:**  Eq. (6) introduces a fusion factor. How does it work? Is it a learnable parameter or a handcraft parameter? If it is a handcraft parameter, how you get the value?
> >
>
> **A2:** In Eq. (6), the introduced fusion factor $\eta$ is used to control the weight of the residual branch in the final feature aggregation. Specifically, Eq. (6) adopts a residual structure that combines the main branch features and the auxiliary branch features through weighted summation, in order to balance their contributions to the final representation.
> $\eta$ is not a learnable parameter; instead, it is a manually defined constant based on empirical evaluation. During model tuning, we conducted systematic experiments with different values of $\eta$, and selected $\eta$=0.2 based on the performance on the validation set. We found that this value effectively regularizes the learning process by leveraging the benefits of residual connections while maintaining feature stability. We chose to perform the ablation study of $η$ only on the CARGO-ALL dataset because it provides a comprehensive evaluation across the entire CARGO benchmark. Therefore, the selection of $\eta$ was based on its performance on CARGO-ALL, which reflects the overall capability of the model across diverse cross-view scenarios.
>
> Table G: Performance comparison of different $\eta$ values on the CARGO-ALL dataset.
>
> | CARGO-ALL | Rank1 | mAP | mINP |
> | --- | --- | --- | --- |
> | $\eta$=0.01 | 64.10 | 57.85 | 44.76 |
> | $\eta$=0.1 | 65.12 | 56.35 | 43.62 |
> | $\eta$=0.2 | 66.67 | 56.35 | 41.42 |
> | $\eta$=0.5 | 64.55 | 56.92 | 44.03 |
>
> > **Q3:** Compared to the original TPS, the proposed LTPS makes some rules during the training. So, how does this change influence the performance when compared to the original TPS?
> >
>
> **A3:** We conducted experiments to compare the performance of the original TPS (i.e., applying original TPS transformations at every layer) with our proposed LTPS. The comparison was conducted on the CARGO dataset, and the results are presented in Table H.
> The results show that while the original TPS can partially alleviate feature misalignment under the aerial-to-ground (A→G) scenario, its fixed rotation and interpolation strategies perform poorly when adapting to ground-to-ground (G→G) view matching. For instance, compared to ViT and VDT, dca-tps achieves a slight improvement under A→G but suffers from degraded performance under G→G, with an overall average performance even lower than ViT.
> In contrast, our proposed LTPS introduces learnable control points and constrained rotation magnitudes through regularized design, which avoids excessive perturbation to local geometric structures. This enables LTPS to maintain strong adaptability under A→G conditions while significantly improving retrieval accuracy under G→G scenarios. Overall, LTPS provides more balanced performance across both view settings, making it a more suitable solution for the task than the standard TPS approach.
>
> Table H: Performance comparison of original TPS and LTPS on the CARGO dataset. Here, GSAlign-O is  ViT applying original TPS transformations at every layer.
>
> | Method | ALL |  |  | G→G |  |  | A→A |  |  | A→G |  |  |
> | --- | --- | --- | --- | --- | --- | --- | --- | --- | --- | --- | --- | --- |
> |  | Rank1 | mAP | mINP | Rank1 | mAP | mINP | Rank1 | mAP | mINP | Rank1 | mAP | mINP |
> | GSAlign-O | 63.78 | 55.18 | 41.63 | 80.36 | 71.21 | 57.06 | 82.50 | 65.44 | 47.37 | 58.51 | 57.72 | 49.02 |
> | GSAlign | 66.67 | 56.35 | 41.42 | 83.04 | 71.57 | 58.68 | 82.50 | 67.70 | 51.80 | 65.96 | 60.50 | 49.32 |

---

> > ### Comment · Reviewer_5Ks3 · 2025-08-03
> >
> > Thank you to the authors for their responses. The experiment on the AG-ReiD dataset and hyperparameters has addressed some of my concerns. I have decided to maintain this positive score.

---

> ### Author Response · Authors · 2025-08-01
>
> We would like to clarify that the model referred to as **dca-tps** in the A3 paragraph above corresponds to **GSAlign-O** in Table H. This inconsistency stems from the early naming used in our internal experimental logs, where “DCA” was originally adopted. We apologize for the oversight and any confusion it may have caused. The results and conclusions remain valid, and the model name should be referred to as GSAlign-O throughout.

---

### Official Review · Reviewer_nnjx · 2025-07-01

**Clarity:** 2
**Significance:** 3
**Originality:** 2
**Rating:** 4
**Confidence:** 3

**Summary:**

This paper proposes a novel framework, called GSAlign, for Aerial-Ground person re-identification (AG-ReID). It effectively integrates a Learnable Thin Plate Spline (LTPS) Module for geometric variations and a Dynamic Alignment Module (DAM) for visibility-aware semantic alignment, achieving significantly improvement on the CARGO dataset in aerial-ground settings.

**Questions:**

Main Concern (If the authors can convince me, I will upgrade the rating):

1. Why does LTPS automatically rotate images correctly? From my perspective, the deformation loss (L_{deform} in Eq. 10) seems to suppress any rotation, and the paper does not seem to use any correctly aligned image as ground truth. Please clarify this. And I also want to know the proportion of correctly rotated images/features.

2. Why is LTPS necessary at every layer? Why does adding it to shallower layers result in poor performance, while adding it to deeper layers yield significant improvements? The authors mention that shallow layers cannot predict rotation well in Line 318. Could this phenomenon be further elucidated through visualizations of the LTPS-processed results at each layer?

3. Could you compare the performance of your method with a wider range of alignment approaches, including implicit methods such as BAU [1], SEAS [2] and PAT [3], as well as explicit alignment techniques such as DAGait [4] and so on? I want to know if LTPS and DAM truly outperform other traditional alignment methods.

4. Could you provide the performance on the AG-ReID [5] dataset? This paper validates its performance on the CARGO dataset, which consists of synthetic data. However, in AG-ReID task, we have an AG-ReID [5] dataset with real data. Can the authors validate the feasibility of their method on real data?

[1] Cho, Y., Kim, J., Kim, W. J., Jung, J., & Yoon, S. E. (2024). Generalizable person re-identification via balancing alignment and uniformity. Advances in Neural Information Processing Systems, 37, 47069-47093.

[2] Zhu, H., Budhwant, P., Zheng, Z., & Nevatia, R. (2024). Seas: Shape-aligned supervision for person re-identification. In Proceedings of the IEEE/CVF Conference on Computer Vision and Pattern Recognition (pp. 164-174).

[3] Ni, H., Li, Y., Gao, L., Shen, H. T., & Song, J. (2023). Part-aware transformer for generalizable person re-identification. In Proceedings of the IEEE/CVF international conference on computer vision (pp. 11280-11289).

[4] Wu, Z., Zhang, C., Xu, H., Jiao, P., & Wang, H. (2025). DAGait: Generalized Skeleton-Guided Data Alignment for Gait Recognition. arXiv preprint arXiv:2503.18830.

[5] Nguyen, H., Nguyen, K., Sridharan, S., & Fookes, C. (2023, July). Aerial-ground person re-id. In 2023 IEEE International Conference on Multimedia and Expo (ICME) (pp. 2585-2590). IEEE.

**Ethical Concerns:**

["NO or VERY MINOR ethics concerns only"]

**Final Justification:**

My concerns have been addressed. Overall I think it is a good appearance / semantic auto-alignment method for AG-ReID, offering some new insights and a good performance improvement in most cases.

**Limitations:**

yes

**Quality:**

2

**Strengths And Weaknesses:**

Quality: This paper is technically sound and presents a complete, appropriate work. Some claims are well-supported by experimental results on CARGO dataset. The authors honestly evaluate the limitations in the Supplementary Material (Sec. A.2).

Clarity: The organization of this submission is fair, but its clarity needs to be improved.

1. In Line 17, "Ecomprehensive" -> "A comprehensive".
2. In Line 86, "Aerial-Ground ..." is not a sentence.
3. In Line 129, "contro" -> "control".
4. In Line 154, "a tunable fusion factors" -> "a tunable fusion factor".
5. This paper builds on traditional thin plate spline (TPS) technique, but it lacks an introduction to TPS and related references in the 'Related Work' section. For better clarity, the authors are expected to provide more related works.
6. LTPS is one of your main contributions, but its details are difficult to comprehend. The authors need rephrase Lines 128-156 (e.g., What is "X" of "X - P^{(rot)}_{s,i} "in Eq. 4? ) and incorporate additional figures to more clearly illustrate how control points are transformed from P_{s} to P_{t}.

Significance: The results of this paper have some impact on the community, moderately advancing our understanding. Others may build upon this idea. However, the submission needs more experiments to convince me that it is a better approach than previous work. For example, the paper mentions traditional hand-crafted alignment strategies in Line 51, but a comparison to them is lacking.

Originality: The contributions of this paper differ from previous works. This paper need to provide more related references about TPS. The combination of LTPS and DAM appears to be a reasonable approach, as both modules aim to address the problem of spatial and semantic misalignment.

---

> ### Author Rebuttal · Authors · 2025-07-31
>
> Thanks for your constructive feedback. Below, we respond to your key concerns point by point.
> > **Q1:**  Why does LTPS automatically rotate images correctly? From my perspective, the deformation loss (L_{deform} in Eq. 10) seems to suppress any rotation, and the paper does not seem to use any correctly aligned image as ground truth. Please clarify this. And I also want to know the proportion of correctly rotated images/features.
> >
>
> **A1:**  Thanks for your suggestion. We would like to clarify that the goal of LTPS is not to generate an "absolutely correct" rotated image, nor does it rely on any pre-aligned ground truth images. LTPS is optimized by the ReID losses to get better alignment. This is the same way with the Spatial Transformation Network [1], which is learned by the optimization of classification losses.
> For a ViT-based model, there is no strictly defined "correct pose" for input images. Instead, the key objective is to alleviate the misalignment between aerial and ground viewpoints through viewpoint adjustment, thereby enabling more stable and discriminative feature representations.
>
> The deformation loss L_deform in Eq. (10) is not intended to suppress rotation entirely. Rather, it serves as a regularization term to prevent excessive geometric deformation (e.g., overly large rotation angles) that could harm feature learning. In our preliminary experiments, directly applying fixed-angle rotations without constraint led to an average drop of about 6% in Rank-1 accuracy (as shown in Table C). Therefore, LTPS aims to mitigate cross-view misalignment through moderate and controlled geometric transformations, rather than pursuing perfect alignment on a per-image basis.
>
> Table C: Performance comparison of fixed-angle rotations and LTPS on the CARGO dataset.
>
> | Method | ALL |  |  | G→G |  |  | A→A |  |  | A→G |  |  |
> | --- | --- | --- | --- | --- | --- | --- | --- | --- | --- | --- | --- | --- |
> |  | Rank1 | mAP | mINP | Rank1 | mAP | mINP | Rank1 | mAP | mINP | Rank1 | mAP | mINP |
> | without L_deform | 60.65 | 52.06 | 38.83 | 78.89 | 68.21 | 57.64 | 77.50 | 59.54 | 40.73 | 58.45 | 53.74 | 43.84 |
> | with L_deform | **66.67** | **56.35** | **41.42** | **83.04** | **71.57** | **57.68** | **82.50** | **67.70** | **51.80** | **65.96** | **60.60** | **49.32** |
>
> ---
>
> > **Q2:** Why is LTPS necessary at every layer? Why does adding it to shallower layers result in poor performance, while adding it to deeper layers yield significant improvements? The authors mention that shallow layers cannot predict rotation well in Line 318. Could this phenomenon be further elucidated through visualizations of the LTPS-processed results at each layer?
> >
>
> **A2:**  As shown in Table 4 in mamuscript, we provide detailed ablation studies to demonstrate the effect of applying LTPS at different network depths. The results indicate that applying LTPS only to shallow or deep layers leads to performance degradation, while applying it across all layers yields the best performance. This is because shallow features are more sensitive to local structures and low-level textures, and using LTPS alone at these layers can be affected by unstable rotation predictions. Deep layers, on the other hand, capture more semantic information, and applying LTPS only at this stage is insufficient to bridge cross-view differences. Applying LTPS across all layers helps maximize the mitigation of viewpoint misalignment. Regarding the reviewer’s concern that "shallow layers cannot predict rotation well",  as NeurIPS rebuttal rules do not allow adding new figures, we are unable to include them at this stage. But we are willing to add them in the camera-ready for a better explanation.
>
> ---
>
> > **Q3:** Could you compare the performance of your method with a wider range of alignment approaches, including implicit methods such as BAU [1], SEAS [2] and PAT [3], as well as explicit alignment techniques such as DAGait [4] and so on? I want to know if LTPS and DAM truly outperform other traditional alignment methods.
> >
>
> **A3:** Following your suggestion, in Table D, we compare various alignment methods, including implicit approaches such as BAU, PAT, and DAGait, as well as the explicit alignment method GSAlign. The results show that BAU[1], PAT[3], and DAGait[4] all perform significantly worse than our baseline and GSAlign on the CARGO dataset. In contrast, our proposed combination of LTPS and DAM consistently achieves superior performance under the same experimental conditions. Notably, DAGait, which is originally designed as an explicit alignment method for gait recognition, performs poorly on person ReID datasets. Its accuracy on the test set is only around 0.6%, which is substantially lower than all other compared methods. Detailed results are provided in Table D for reference. As for SEAS[2], since the official code has not been released and the limited rebuttal period, we were unable to include its comparison in time.
>
> Table D: Performance comparison of the implicit methods and alignment method under four settings of the CARGO dataset.
>
> | Method | ALL |  | G→G |  | A→A |  | A→G |  |
> | --- | --- | --- | --- | --- | --- | --- | --- | --- |
> |  | Rank1 | mAP | Rank1 | mAP | Rank1 | mAP | Rank1 | mAP |
> | BAU [1] | 45.20 | 38.40 | 61.60 | 51.20 | 50.00 | 42.60 | 40.40 | 36.70 |
> | PAT [3] | 37.90 | 15.30 | 52.70 | 24.20 | 50.00 | 23.10 | 35.10 | 15.50 |
> | DAGait [4] | 0.66 | - | 0.77 | - | 0.64 | - | 0.42 | - |
> | GSALign | **66.67** | **56.35** | **83.04** | **71.57** | **82.05** | **67.70** | **65.96** | **60.50** |
>
> [1] Cho, Yoonki, et al. "Generalizable person re-identification via balancing alignment and uniformity." NeurlPS, 2024.
>
> [2] Zhu, Haidong, et al. "Seas: Shape-aligned supervision for person re-identification." CVPR, 2024. seas
>
> [3] Ni, Hao, et al. "Part-aware transformer for generalizable person re-identification." CVPR,  2023. Pat
>
> [4] Wu, Z., et al. “AGait: Generalized Skeleton-Guided Data Alignment for Gait Recognition.” ArXiv, 2025
>
> > **Q4:** Could you provide the performance on the AG-ReID dataset? This paper validates its performance on the CARGO dataset, which consists of synthetic data. However, in AG-ReID task, we have an AG-ReID dataset with real data. Can the authors validate the feasibility of their method on real data?
> >
>
> **A4:** Following your suggestion, we conducted comprehensive comparative experiments on the AG-ReID dataset [1], covering both cross-domain settings: A→G and G→A. As shown in Table E, extensive experiments on this challenging real-world dataset demonstrate the effectiveness of GSAlign. Compared to previous state-of-the-art methods, our proposed GSAlign still achieves significant improvements.
>
> Table E: Performance comparison of the mainstream methods under two settings of the AG-ReID dataset. Here, VDT-R is the reproduced version of its open-source code.
>
> | Method | A→G |  |  | G→A |  |  |
> | --- | --- | --- | --- | --- | --- | --- |
> |  | Rank1 | mAP | mINP | Rank1 | mAP | mINP |
> | SBS [6] | 73.54 | 59.77 | - | 73.70 | 62.27 | - |
> | BoT [7] | 70.01 | 55.47 | - | 71.20 | 58.83 | - |
> | OsNet [8] | 72.59 | 58.32 | - | 74.22 | 60.99 | - |
> | VV [9] | 77.22 | 67.23 | 41.43 | 79.73 | 69.83 | 42.37 |
> | AG-ReIDv1 [5] | 81.28 | 72.38 | - | 82.64 | 73.35 | - |
> | ViT [10] | 81.47 | 72.61 | - | 82/85 | 73.39 | - |
> | VDT [11] | 82.91 | 74.44 | 51.06 | 86.59 | 78.57 | 52.87 |
> | VDT-R [11] | 82.73 | 74.30 | 51.77 | 84.62 | 76.56 | 49.70 |
> | GSAlign | **86.74** | **84.00** | **73.62** | **87.94** | **87.17** | **78.21** |
>
> [5] Nguyen, Huy, et al. "Aerial-ground person re-id." ICME, 2023.
>
> [6] He, Lingxiao, et al. "Fastreid: A pytorch toolbox for general instance re-identification." MM, 2023.
>
> [7] Luo, Hao, et al. "Bag of tricks and a strong baseline for deep person re-identification." CVPR, 2019.
>
> [8] Zhou, Kaiyang, et al. "Learning generalisable omni-scale representations for person re-identification." TPAMI, 2021.
>
> [9] Kuma, Ratnesh, et al. "Vehicle re-identification: an efficient baseline using triplet embedding." IJCNN, 2019.
>
> [10] Dosovitskiy, Alexey, et al. "An image is worth 16x16 words: Transformers for image recognition at scale." ICLR, 2021
>
> [11] Zhang, Quan, et al. "View-decoupled transformer for person re-identification under aerial-ground camera network." CVPR, 2024.

---

> > ### Comment · Reviewer_nnjx · 2025-08-01
> >
> > Thanks for your constructive feedback.
> > ### **Q1:**
> > - This question has been solved in some extend by providing related work about Spatial Transformation Network [1], even though its reference link is missing. As mentioned in the Clarity of "Strengths And Weaknesses", this paper builds on traditional thin plate spline (TPS) technique, but it heavily lacks an introduction to TPS and related works. For better understanding and positioning clearer, the "Related Work" section are expected to rewrite.
> >
> > - As you mentioned in"In our preliminary experiments, directly applying fixed-angle rotations without constraint led to an average drop ....", why without constraint (L_deform loss) binds with a fixed-angle rotations? Does the "without L_deform" row in Tab. C use a fixed rotation angle, instead of allowing it to be freely optimized under the supervision of the ReID loss? In contrast, does the "with L_deform" setting enable a flexible rotation angle that is jointly optimized by both the ReID loss and L_deform? Or do I have a misunderstanding?
> >
> > [1] Jaderberg, Max, Karen Simonyan, and Andrew Zisserman. "Spatial transformer networks." Advances in neural information processing systems 28 (2015).
> >
> > ### **Q2:**
> > - Although the response on the function of layer-wise LTPS still lacks convincing data and visualizations, I will pause this question, as NeurIPS rebuttal rules do not allow adding new figures. The visualization analysis is encourage to update on the manuscript.
> >
> > ### **Q3:**
> > - Why the explicit alignment DAGait obtain such low performance? The data seems unreliable. Do you use a similar, common, and comparable backbone between DAGait and GSAlign for feature extraction? This experiments just want to make sure that you have a better alignment method than others, so you should make the backbone comparable except the alignment part.
> > - Thanks for your detailed data. This table is encouraged to update on the manuscript to highlight your contribution on alignment methods.
> >
> > ### **Q4:**
> > - Thanks for your results. The performance on AG-ReID and AG-ReIDv2 looks good. And this result should be updated on your final manuscript.

---

> ### Author Response · Authors · 2025-08-02
>
> > **Q1:**
> >
> > - This question has been solved in some extend by providing related work about Spatial Transformation Network [1], even though its reference link is missing. As mentioned in the Clarity of "Strengths And Weaknesses", this paper builds on traditional thin plate spline (TPS) technique, but it heavily lacks an introduction to TPS and related works. For better understanding and positioning clearer, the "Related Work" section are expected to rewrite.
> > - As you mentioned in"In our preliminary experiments, directly applying fixed-angle rotations without constraint led to an average drop ....", why without constraint (L_deform loss) binds with a fixed-angle rotations? Does the "without L_deform" row in Tab. C use a fixed rotation angle, instead of allowing it to be freely optimized under the supervision of the ReID loss? In contrast, does the "with L_deform" setting enable a flexible rotation angle that is jointly optimized by both the ReID loss and L_deform? Or do I have a misunderstanding?
>
>
>
> A1:
>
> - Thank you for the suggestion. We will include an introduction and proper citation of the TPS technique in the revised version. The "Related Work" section will also be reorganized to better position our method and clarify its connection to prior work such as Spatial Transformer Networks [1].
> - Thank you for the detailed question. In our preliminary results (Table C in the rebuttal), we used fixed-angle rotations to demonstrate that large, unconstrained transformations can hurt performance. This motivated the introduction of the L_deform constraint in our method design. To address your concern, we conducted additional experiments where the rotation angle is freely optimized under the supervision of the ReID loss, without using L_deform. The results (shown in Table J) further confirm the necessity of L_deform, as its inclusion consistently improves performance. To avoid potential misunderstandings, we have updated the table accordingly:  **"LTPS without L_deform"** refers to the setting where the rotation angle is **freely optimized under the supervision of the ReID loss alone**, without any deformation constraint. **"Fixed-angle rotation"** corresponds to the original **"without L_deform"** setting in Table C, where a fixed rotation angle is applied. **"L_deform"** denotes our final proposed scheme, where the rotation angle is jointly optimized under both the ReID loss and the L_deform constraint.
>
> Table J: Performance comparison of fixed-angle rotation and LTPS on the CARGO dataset.
>
> | Method | ALL |  |  | G→G |  |  | A→A |  |  | A→G |  |  |
> | --- | --- | --- | --- | --- | --- | --- | --- | --- | --- | --- | --- | --- |
> |  | Rank1 | mAP | mINP | Rank1 | mAP | mINP | Rank1 | mAP | mINP | Rank1 | mAP | mINP |
> | LTPS without  L_deform | 59.94 | 52.67 | 39.25 | 76.79 | 69.83 | 57.68 | 75.00  | 43.49  | 43.49 | 56.38 | 53.22 | 43.96 |
> | fixed-angle rotation | 60.65 | 52.06 | 38.83 | 78.89 | 68.21 | 57.64 | 77.50 | 59.54 | 40.73 | 58.45 | 53.74 | 43.84 |
> | with L_deform | **66.67** | **56.35** | **41.42** | **83.04** | **71.57** | **57.68** | **82.50** | **67.70** | **51.80** | **65.96** | **60.60** | **49.32** |
>
> [1] Jaderberg, Max, Karen Simonyan, and Andrew Zisserman. "Spatial transformer networks." Advances in neural information processing systems 28 (2015).

---

> ### Author Response · Authors · 2025-08-02
>
> > **Q2:**
> >
> > - Although the response on the function of layer-wise LTPS still lacks convincing data and visualizations, I will pause this question, as NeurIPS rebuttal rules do not allow adding new figures. The visualization analysis is encourage to update on the manuscript.
>
> **A2:** Thank you for your understanding. This is a valuable suggestion. We will include the visualization results and related discussion in the final manuscript as suggested.
>
> > **Q3:**
> >
> > - Why the explicit alignment DAGait obtain such low performance? The data seems unreliable. Do you use a similar, common, and comparable backbone between DAGait and GSAlign for feature extraction? This experiments just want to make sure that you have a better alignment method than others, so you should make the backbone comparable except the alignment part.
> > - Thanks for your detailed data. This table is encouraged to update on the manuscript to highlight your contribution on alignment methods.
>
> **A3:**
>
> - Thank you for the question. DAGait is designed to work with datasets that provide skeleton annotations, which are essential for its alignment process. However, the CARGO dataset does not include such annotations. Although we applied a similar feature extraction process when adapting DAGait to CARGO, the absence of skeleton labels prevents it from functioning effectively, leading to lower performance. To further address your concern, we additionally implemented another explicit alignment method, DTST [2]. As shown in Table K, our proposed method consistently outperforms DTST across most settings. Notably, GSAlign achieves a significant improvement in the challenging A→G case.
>
> Table K: Performance comparison of alignment method and GSAlign on the CARGO dataset.
>
> | Method | ALL |  |  | G→G |  |  | A→A |  |  | A→G |  |  |
> | --- | --- | --- | --- | --- | --- | --- | --- | --- | --- | --- | --- | --- |
> |  | Rank1 | mAP | mINP | Rank1 | mAP | mINP | Rank1 | mAP | mINP | Rank1 | mAP | mINP |
> | DTST [2] | 64.42 | 55.73 | **41.92** | 78.57 | **72.40** | **62.10** | 80.00 | 63.31 | 44.67 | 50.53 | 43.49 | 29.46 |
> | GSAlign | **66.67** | **56.35** | 41.42 | **83.04** | 71.57 | 57.68 | **82.50** | **67.70** | **51.80** | **65.96** | **60.60** | **49.32** |
>
> [2] Wang, Yuhai, et al. "Dynamic Token Selection for Aerial-Ground Person Re-Identification." ICME, 2025
>
> > **Q4:**
> >
> > - Thanks for your results. The performance on AG-ReID and AG-ReIDv2 looks good. And this result should be updated on your final manuscript.
> >
>
> **A4:** Thank you for recognizing our experimental results. We will include the additional experiments in the final version of the manuscript as suggested.

---

> > ### Comment · Reviewer_nnjx · 2025-08-04
> >
> > Thanks for your response. My concerns have been well addressed. I will improve my score.

---

> > > ### Author Response · Authors · 2025-08-04
> > >
> > > We sincerely appreciate your positive feedback. We are pleased that our response has addressed your concerns, and we are grateful for your recognition and support of our work.

---

### Official Review · Reviewer_tvZF · 2025-07-02

**Clarity:** 2
**Significance:** 2
**Originality:** 2
**Rating:** 3
**Confidence:** 3

**Summary:**

The paper integrates a learnable thin plate spline module (LTPS) and a dynamic alignment module (DAM) into the view-decoupled transformer (VDT) framework [1] for aerial-ground person re-identification. It demonstrates that the proposed method (GSAlign) outperforms existing methods, including VDT, on the CARGO dataset [1] across most setups.

[1] Zhang, Quan, et al. "View-decoupled transformer for person re-identification under aerial-ground camera network." Proceedings of the IEEE/CVF Conference on Computer Vision and Pattern Recognition. 2024.

**Questions:**

1. Please include experimental results on real-world aerial-ground person re-identification datasets, such as AG-ReID [2] and its successor, AG-ReID v2 [3].

2. Please provide an explanation for why the proposed LTPS and DAM modules lead to decreased performance in the aerial-aerial scenarios.

3. Please improve the overall clarity of the paper according to the issues outlined in Weakness 3.

**Ethical Concerns:**

["NO or VERY MINOR ethics concerns only"]

**Final Justification:**

Although my concerns regarding the aerial-to-aerial (A $\rightarrow$ A) scenarios remain, the additional results on the AG-ReID and AG-ReID v2 datasets provided by the authors clearly strengthen the proposed GSAlign method. As a result, I have decided to slightly raise my assessment of the paper.

**Limitations:**

Yes.

**Quality:**

2

**Strengths And Weaknesses:**

Strengths:

1. The concept of explicitly modeling geometric alignment (to handle viewpoint variances) and semantic alignment (to address foreground and background issues) within the ReID network presents a promising solution that directly tackles the core challenges of aerial-ground ReID.

2. The proposed GSAlign method outperforms existing approaches, such as VDT, on the CARGO dataset across most setups, particularly in the targeted aerial-ground scenarios.

Major Weaknesses:

1. Missing results on major real-world benchmarks: The paper demonstrates GSAlign’s superior performance solely on the CARGO dataset [1], which is a synthetic benchmark for aerial-ground person ReID. However, it does not evaluate the method’s effectiveness on real-world datasets such as AG-ReID [2] and its successor, AG-ReID v2 [3]. Notably, the AG-ReID dataset was also used by VDT [1], the method upon which GSAlign is built, for performance evaluation. Assessing GSAlign on these real-world datasets is essential to determine its practical applicability and generalizability beyond synthetic environments.

2. Accuracy drops in aerial-aerial scenarios: Although GSAlign outperforms VDT in most configurations, particularly in the aerial-ground scenarios, it underperforms in the aerial-aerial scenarios. As shown in Tab. 2, this performance drop is attributed to both the LTPS and DAM modules. If, as stated in Ln 266, the aerial-ground setting is indeed “the most challenging scenario”, it raises the question of why the proposed modules negatively impact the supposedly easier aerial-aerial scenarios. This inconsistency is not addressed in the paper.

Minor Weaknesses:

3. The overall quality and clarity of the paper can be improved in several respects:

- While numerous prior works have explored integrating Thin Plate Spline (TPS) into learning frameworks [4]–[6], the paper does not reference any of them. It remains unclear what the major technical contribution of the proposed LTPS module is in comparison to these earlier approaches.

- The citation for VDT in Ln 103 appears to be incorrect and should be verified.

- The clarity of Fig. 2 could be significantly improved. Terms such as “input tokens”, “located points”, and “aligned tokens” are shown in the diagram but are not defined or used elsewhere in the text, making the illustration difficult to interpret.

- The paper does not clearly explain the distinction between the proposed DAM module and the existing DPM method. Clarifying this difference is crucial to justify DAM's novelty.

- The rationale for discussing methods like Memory Bank and Classification Matrix in Tab. 5 is unclear. Are these methods derived from or related to prior work? Their inclusion needs further justification.

[2] Nguyen, Huy, et al. "Aerial-ground person re-id." 2023 IEEE International Conference on Multimedia and Expo (ICME). IEEE, 2023.

[3] Nguyen, Huy, et al. "AG-ReID. v2: Bridging aerial and ground views for person re-identification." IEEE Transactions on Information Forensics and Security 19 (2024): 2896-2908.

[4] Jaderberg, Max, Karen Simonyan, and Andrew Zisserman. "Spatial transformer networks." Advances in neural information processing systems 28 (2015).

[5] Zhao, Jian, and Hui Zhang. "Thin-plate spline motion model for image animation." Proceedings of the IEEE/CVF Conference on Computer Vision and Pattern Recognition. 2022.

[6] Nie, Lang, et al. "Semi-supervised coupled thin-plate spline model for rotation correction and beyond." IEEE Transactions on Pattern Analysis and Machine Intelligence (2024).

[7] Tan, Lei, et al. "Dynamic prototype mask for occluded person re-identification." Proceedings of the 30th ACM international conference on multimedia. 2022.

---

> ### Author Rebuttal · Authors · 2025-07-31
>
> Thanks for your constructive feedback. Below, we respond to your key concerns point by point.
> > **Q1:**  Please include experimental results on real-world aerial-ground person re-identification datasets, such as AG-ReID and its successor, AG-ReID v2.
> >
>
> **A1:** Following your suggestion, we conducted comprehensive comparative experiments on the AG-ReID [1] and AG-ReID v2 [2] datasets. Compared to previous state-of-the-art methods, our proposed GSAlign still achieves significant improvements. These new results will be included in the final version to strengthen the empirical validation of the proposed GSAlign further.
>
> Table A: Performance comparison of the mainstream methods under two settings of the AG-ReID dataset. Here, VDT-R is the reproduced version of its open-source code.
>
> | Method | A→G |  |  | G→A |  |  |
> | --- | --- | --- | --- | --- | --- | --- |
> |  | Rank1 | mAP | mINP | Rank1 | mAP | mINP |
> | SBS [3] | 73.54 | 59.77 | - | 73.70 | 62.27 | - |
> | BoT [4] | 70.01 | 55.47 | - | 71.20 | 58.83 | - |
> | OsNet [5] | 72.59 | 58.32 | - | 74.22 | 60.99 | - |
> | VV [6] | 77.22 | 67.23 | 41.43 | 79.73 | 69.83 | 42.37 |
> |  AG-ReIDv1 [1] | 81.28 | 72.38 | - | 82.64 | 73.35 | - |
> | ViT [7] | 81.47 | 72.61 | - | 82/85 | 73.39 | - |
> | VDT [8] | 82.91 | 74.44 | 51.06 | 86.59 | 78.57 | 52.87 |
> | VDT-R [8] | 82.73 | 74.30 | 51.77 | 84.62 | 76.56 | 49.70 |
> | GSAlign | **86.74** | **84.00** | **73.62** | **87.94** | **87.17** | **78.21** |
>
> Table B: Performance comparison of the mainstream methods under four settings of the AG-ReID v2 dataset
>
> | Method | A→G |  | G→A |  | A→W |  | W→A |  |
> | --- | --- | --- | --- | --- | --- | --- | --- | --- |
> |  | Rank1 | mAP | Rank1 | mAP | Rank1 | mAP | Rank1 | mAP |
> | BoT [4] | 85.40 | 77.03 | 84.65 | 75.90 | 89.77 | 80.48 | 84.65 | 76.90 |
> | AG-ReIDv1 [1] | 87.70 | 79.00 | 87.35 | 78.24 | 93.67 | 83.14 | 87.73 | 79.08 |
> | VDT [8] | 86.46 | 79.13 | 86.14 | 78.12 | 90.00 | 82.21 | 85.26 | 78.52 |
> | AG-ReIDv2 [2] | 88.77 | 80.72 | 87.86 | 78.51 | **93.62** | 84.85 | **88.61** | 80.11 |
> | SeCap [9] | 88.12 | 80.84 | 88.24 | 79.99 | 91.44 | 84.01 | 87.56 | 80.15 |
> | GSALign | **91.47** | **89.78** | **88.29** | **87.62** | 93.30 | **91.84** | 88.12 | **88.62** |
>
> [1]Nguyen, Huy, et al. "Aerial-ground person re-id." ICME, 2023.
>
> [2]Nguyen, Huy, et al. "AG-ReID. v2: Bridging aerial and ground views for person re-identification." IEEE TIFS, 2024.
>
> [3]He, Lingxiao, et al. "Fastreid: A pytorch toolbox for general instance re-identification." MM, 2023.
>
> [4]Luo, Hao, et al. "Bag of tricks and a strong baseline for deep person re-identification." CVPR,  2019.
>
> [5]Zhou, Kaiyang, et al. "Learning generalisable omni-scale representations for person re-identification." TPAMI, 2021.
>
> [6]Kuma, Ratnesh, et al. "Vehicle re-identification: an efficient baseline using triplet embedding." IJCNN, 2019.
>
> [7]Dosovitskiy, Alexey, et al. "An image is worth 16x16 words: Transformers for image recognition at scale."  ICLR, 2021
>
> [8]Zhang, Quan, et al. "View-decoupled transformer for person re-identification under aerial-ground camera network." CVPR, 2024.
>
> [9] Wang, Shining, et al. "SeCap: Self-Calibrating and Adaptive Prompts for Cross-view Person Re-Identification in Aerial-Ground Networks." CVPR,  2025.
>
>
> > **Q2:** Please provide an explanation for why the proposed LTPS and DAM modules lead to decreased performance in the aerial-aerial scenarios.
> >
>
> **A2:**  Thanks for your suggestion. This work is mainly focused on the viewpoint variations in the aerial-to-ground (A→G) setting and reaches a satisfying performance. Although the aerial-to-aerial(A→A) scenario involves similar viewpoints, pedestrian targets are typically very small, with scarce appearance textures and relatively uniform backgrounds. In such cases, the model does not need extra alignment transformations. Our proposed LTPS and DAM modules are designed to handle large pose variations and spatial misalignments, which are particularly severe in the aerial-to-ground (A→G) setting. These modules significantly improve performance in A→G scenarios. However, in A→A settings where geometric differences are limited, such modules may introduce unnecessary local transformations, resulting in a slight performance drop. Overall,  our method still achieves competitive results in the A→A scenario and delivers substantial improvements in the core task of A→G re-identification, demonstrating its overall effectiveness and robustness. Your suggestion indeed points us toward future research directions in which we can explore in the future: how to improve aerial-to-ground (A→G)  performance while maintaining aerial-to-aerial(A→A) performance. We will include this discussion in the limitation section. Thank you again for your constructive feedback.
>
> > **Q3:** Please improve the overall clarity of the paper according to the issues outlined in Weakness 3.
> >
>
> **A3:** We thank the reviewer for the constructive comments and helpful suggestions to improve our paper.
>
> - We would like to clarify that LTPS is fundamentally different from traditional TPS-based approaches. While prior works typically apply TPS as a fixed pre-processing step or uniformly across network layers, our LTPS module is designed as a task-specific, learnable transformation mechanism that is selectively activated within a ViT-based ReID framework. Unlike classical TPS, LTPS is optimized directly by the ReID loss to adaptively adjust spatial representations and mitigate aerial-ground viewpoint misalignment within a ViT structure. Moreover, LTPS introduces a deformation loss to regularize transformation strength, ensuring that geometric changes are meaningful and moderate. This prevents excessive distortion, which we found in preliminary experiments to significantly degrade performance. To our knowledge, this is the first work to incorporate a regularized and partially-activated TPS module into a transformer-based architecture for person ReID. We will revise the manuscript to clearly highlight these distinctions and cite relevant TPS-based studies.
>
> - The incorrect citation of VDT (Ln 103) will be corrected.
>
> - We agree that Fig. 2 can be improved. In the final version, we will refine the figure and define terms such as “input tokens,” “located points,” and “aligned tokens” for clarity.
>
> - The distinction between DAM and DPM will be clarified. Different from DPM, which uses the classification matrix as the prototype and gets the mask from sample-to-prototype similarity, DAM directly uses the inner batch sample-to-sample similarity to achieve semantic alignment. Experimental results in Table 5 show the advantage of DAM (Inner-Batch) when compared to the DPM (Classification Matrix). We will also provide proper context and justification for the use of Memory Bank and Classification Matrix in Table 5.

---

> > ### Comment · Reviewer_tvZF · 2025-08-04
> >
> > First, I would like to thank the authors for addressing my concerns within the short rebuttal period.
> >
> > The additional results on AG-ReID and AG-ReID v2 are promising and demonstrate the applicability of the proposed GSAlign method to real-world scenarios.
> >
> > However, I respectfully disagree with the authors' claim that aerial-to-aerial (A $\rightarrow$ A) scenarios involve similar viewpoints and thus do not require additional transformation alignment. Given the considerable viewpoint diversity that can arise from UAV flights, different aerial perspectives, such as nadir versus oblique views, could still benefit from methods that explicitly handle geometric transformation alignment. The observed accuracy drops of GSAlign raise concerns about whether its performance improvements are genuinely due to effective geometric alignment or rather to other underlying mechanisms.
> >
> > Despite these concerns regarding the A $\rightarrow$ A setting, the additional results on real-world datasets (AG-ReID and AG-ReID v2) certainly strengthen the paper. Accordingly, I will raise my score to 3.

---

> > > ### Author Response · Authors · 2025-08-05
> > >
> > > We are sincerely thankful for your thoughtful comments and for taking the time to review our work. Your recognition is deeply appreciated.

---

> ### Author Response · Authors · 2025-08-04
>
> Dear Reviewer tvZF,
> ﻿
>
> Thanks again for your great efforts and constructive advice in reviewing this paper! As the discussion period progresses, we expect your feedback and thoughts on our reply. We put a significant effort into our response, with several new experiments and discussions. We really hope you'll consider our reply. We look forward to hearing from you, and we can further address unclear explanations and remaining concerns if any.
> ﻿
>
> Best regards,
> ﻿
>
> Authors

---

### Note · Authors · 2025-08-15

Dear Reviewers and AC,

 We are encouraged that all reviewers recognized the novelty of GSAlign, especially the integration of the Learnable Thin Plate Spline (LTPS) and Dynamic Alignment Module (DAM) for aerial–ground person re-identification. Reviewers appreciated the clear motivation, solid baseline design, comprehensive ablations, and competitive performance on the CARGO dataset.

Reviewers recognized its strong capability in geometric and semantic alignment, leading to state-of-the-art performance in aerial–ground person re-identification. We well-addressed most reviewers’ concerns by (i) conducting additional validation on real-world AG-ReID [ICME’23] and AG-ReID v2 [TIFS’24] benchmarks beyond the synthetic CARGO dataset, (ii) providing detailed parameter selection explanations, and (iii) adding comprehensive comparisons with both implicit (BAU, PAT) and explicit (DAGait) alignment methods under identical settings. On AG-ReID, GSAlign achieved +22.56% mINP and +3.83% Rank-1 over VDT, and on AG-ReID v2, +8.94% mAP and +3.35% Rank-1 over the best baseline, confirming its strong generalization to real-world aerial–ground scenarios. One reviewer expressed concern about the slightly lower performance in the A→A setting. We clarified that this scenario involves minimal geometric misalignment, where the LTPS and DAM modules—designed specifically to handle severe aerial–ground geometric and semantic differences—provide limited benefit and may even introduce minor variations. Nevertheless, GSAlign remains competitive in same-view cases while delivering substantial gains in the core and more challenging A→G tasks, which are the main focus of AG-ReID research.

We will explicitly discuss GSAlign’s limitations in small geometric-difference settings, refine figure quality, ensure all terms are clearly defined, and correct minor citation issues. Additional visualizations will be added to better illustrate alignment effects, and dataset details will be clarified for reproducibility.

We sincerely thank the AC and all reviewers for their thorough evaluations and constructive suggestions. Their feedback has led to stronger experiments, clearer methodology descriptions, and an improved manuscript that better demonstrates GSAlign’s robustness, effectiveness, and potential for real-world deployment.
﻿

Best regards!

Authors

---

### Decision · Program_Chairs · 2025-09-17

**Decision:**

Accept (poster)

**Comment:**

The paper proposed a new method for the aerial-ground person re-identification task. It received one accept, two borderline accepts, and one borderline reject from four very professional reviewers. Reviewers acknowledged the novelty/contribution of this paper, and promising experimental results. After rebuttal, all reviewers acknowledged that their concerns had been addressed except that Reviewer tvZF was still not very satisfactory with it but was not against acceptance of the paper.

Reviewer tvZF acknowledged authors' effort in addressing most of his/her concerns, but one remaining concern was the unclear performance drop regarding the aerial-to-aerial (A-A) scenario. Authors provided explanation that this scenario involved minimal geometric misalignment which made their proposed alignment method less effective. However, reviewer tvZF argued that there is considerable viewpoint diversity from UAV flights. AC agreed with this argument, however, different from reviewer tvZF who concerned the general effectiveness of the proposed method, AC thought that it is reasonable for a new method not being able to handle all scenarios perfectly. There may be quite different characteristics in the A-A scenario that the proposed method does not account for. Nevertheless, AC believes that the authors have made a valuable contribution to the field, by providing a novel method which is validated to be effective in different challenging AG scenarios. It could be a remained open question for authors or other researchers to further improve the method for more broader scenarios.

Therefore, considering that all reviewers acknowledged the contribution of the paper and accepted the rebuttal, except the last concern, which was not a critical flaw, AC is happy to accept the paper for publication. Authors are required to update the rebuttal and discussion contents to the camera-ready version of the paper to improve it so as to address the raised concerns in the final paper.